# Small vault RNA1-2 modulates expression of cell membrane proteins through nascent RNA silencing

Adele Alagia[1],*, Jana Tereňová[1],*, Ruth F Ketley[1], Arianna Di Fazio[1], Irina Chelysheva[2], Monika Gullerova[1]

**Gene expression can be regulated by transcriptional or post-transcriptional gene silencing. Recently, we described nuclear nascent RNA silencing that is mediated by Dicer-dependent tRNA-derived small RNA molecules. In addition to tRNA, RNA polymerase III also transcribes vault RNA, a component of the ribonucleoprotein complex vault. Here, we show that Dicer-dependent small vault RNA1-2 (svtRNA1-2) associates with Argonaute 2 (Ago2). Although endogenous vtRNA1-2 is present mostly in the cytoplasm, svtRNA1-2 localises predominantly in the nucleus. Furthermore, in Ago2 and Dicer knockdown cells, a subset of genes that are up-regulated at the nascent level were predicted to be targeted by svtRNA1-2 in the intronic region. Genomic deletion of vtRNA1-2 results in impaired cellular proliferation and the up-regulation of genes associated with cell membrane physiology and cell adhesion. Silencing activity of svtRNA1-2 molecules is dependent on seed-plus-complementary-paired hybridisation features and the presence of a 5-nucleotide loop protrusion on target RNAs. Our data reveal a role of Dicer-dependent svtRNA1-2, possessing unique molecular features, in modulation of the expression of membrane-associated proteins at the nascent RNA level.**

## Introduction

Regulation of gene expression is fundamental for the control of cellular processes such as stemness, differentiation, development, and adaptation to environmental cues (Buccitelli & Selbach, 2020). Control of RNA expression at the transcriptional and post-transcriptional levels can be achieved by several mechanisms (Alagia & Gullerova, 2022). For example, changes in chromatin accessibility through transcriptional gene silencing, and mRNA translation inhibition or degradation by post-transcriptional gene silencing, can efficiently coordinate the silencing of the cellular transcriptome (Martienssen & Moazed, 2015). RNA interference–related gene silencing mechanisms facilitate specific RNA down-regulation with the "guidance" of various classes of small non-coding RNA (sncRNA) molecules such as miRNA (Liu et al, 2018), siRNA, and Piwi-interacting RNA (Moazed, 2012). Furthermore, tRNA-derived small RNA (tsRNA) (Liu et al, 2021), rRNA-derived fragments (Lambert et al, 2019), snoRNA-derived RNAs, and YRNA-derived fragments have been recently identified as small non-coding regulatory RNA molecules (Valkov & Das, 2020).

We have previously identified a class of tsRNA molecules derived from tRNAs, produced by Dicer cleavage activity, and loaded onto Ago2, which target RNA intronic sequences resulting in nascent RNA silencing (NRS) (Di Fazio et al, 2022). The mechanism of NRS differs from transcriptional gene silencing and post-transcriptional gene silencing as the transcription of the target gene remains unaffected and the nascent RNA is degraded within the nucleus. Although we described the fundamental mechanism of NRS, our understanding of the molecular and functional details remains limited.

In the human genome, vault RNAs are located on chromosome 5q31 within two different loci: the VAULT-1 locus, which consists of three genes (VTRNA1-1, VTRNA1-2, and VTRNA1-3), and the VAULT-2 locus, which encodes for VAULT2-1 (Buscher et al, 2020). Both loci are under the control of an RNA polymerase III type 2 promoter, which contains box A and box B motifs (van Zon et al, 2001), typically found within tRNA genes. Vault RNA molecules are 80–140 nucleotides long (Stadler et al, 2009; Ferro et al, 2022) and classified as sncRNAs. vtRNA was initially identified as a component of the large ribonucleoprotein particle known as Vault, which is involved in several cellular processes, such as nuclear transport (Kedersha et al, 1991; Dickenson et al, 2007), immune response (Berger et al, 2009), and drug resistance (Scheffer et al, 1995; Kickhoefer et al, 1998; Mossink et al, 2003). However, it has been shown that most of the vtRNA molecules (~95%) are not associated with the vault particles (Kickhoefer et al, 2002), suggesting that vtRNA molecules might have additional roles within the cell, including as riboregulators of autophagy (Horos et al, 2019) or as effectors of gene expression control in a miRNA-like fashion (Hahne et al, 2021). Indeed, some vtRNA molecules can be processed into smaller fragments called small vault RNA (svtRNA), and their biogenesis can be Drosha- and Dicer-independent (i.e., svtRNA1-1), or Drosha-

---

[1]Sir William Dunn School of Pathology, University of Oxford, Oxford, UK    [2]Oxford Vaccine Group, Department of Paediatrics, University of Oxford, and the NIHR Oxford Biomedical Research Centre, Oxford, UK

Correspondence: monika.gullerova@path.ox.ac.uk
*Adele Alagia and Jana Tereňová contributed equally to this work

---

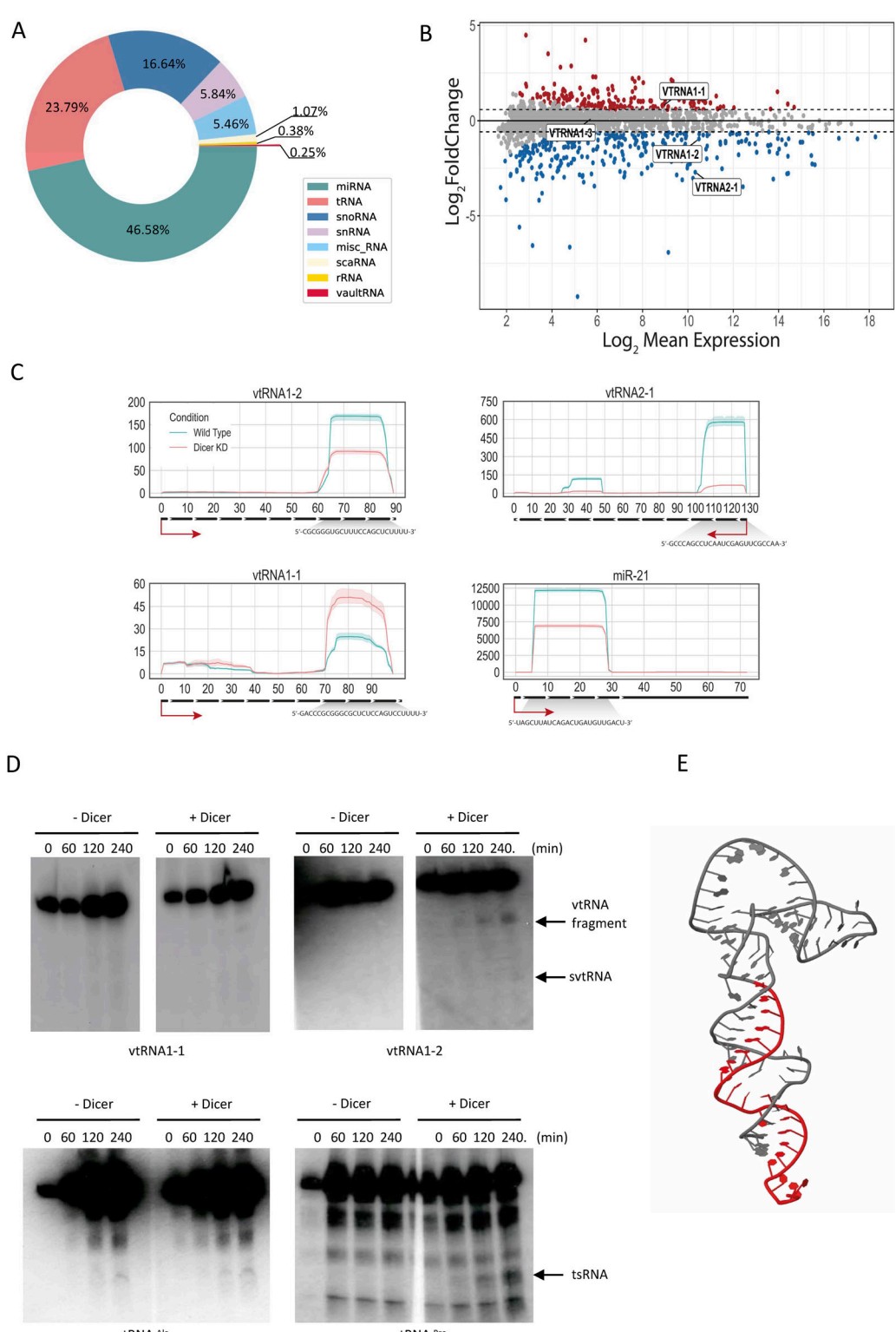

**Figure 1. Small RNA-seq reveals Dicer processing of vtRNA1-2 into smaller RNA fragments.**

**(A)** Pie chart shows the relative abundance of different RNA species with differentially altered expression in WT and Dicer KD sRNA-seq samples. **(B)** MA plot of Dicer KD versus WT. Points in red indicate sRNAs with an adjusted $P < 0.05$, and the points encircled in blue indicate vault RNA. **(C)** Per-base read coverage of vtRNA. The x-axis shows the length of transcripts, and the y-axis shows the normalised number of mapped reads in RPM (Reads Per Million). miR-21 acts as a positive control. **(D)** Representative Northern blot images showing signals of vtRNA1-1, vtRNA1-2, tRNA[Pro], and tRNA[Ala] at different time points after the addition of purified 6xHis-Dicer. Dicer cleavage products are indicated by arrows. **(E)** Predicted tertiary structure of vtRNA1-2. The identified small vault RNA1-2 fragment is coloured in red.

independent and Dicer-dependent (*i.e.* svtRNA2-1) (Hahne et al, 2021). Furthermore, svtRNA biogenesis can be regulated by serine/arginine-rich splicing factor SRSF2 and deposition of 5-methylcytosine modification. svtRNA1-1 and svtRNA2-1 molecules can also associate with Ago proteins, regulate the expression of genes involved in several cellular pathways such as drug metabolism, encode voltage-gated calcium channels, and act as tumour suppressors (Fort et al, 2020).

Here, we show that vtRNA1-2 molecule can be actively processed by Dicer into svtRNA1-2 fragments, associate with Ago2, and localise predominantly in the nucleus. Using the target prediction tool, miRanda, we identified 227 unique svtRNA1-2 targets with dominant intronic complementarity among genes up-regulated in Ago2 and Dicer KD cells. We validated svtRNA1-2–mediated NRS by assessing the nascent and steady-state levels of four predicted target genes. In addition, analysis of chromatin-associated RNA-seq (ChrRNA-seq) in HEK293T vtRNA1-2 knockout cells confirmed the involvement of svtRNA1-2 in NRS of a subset of genes, associated with plasma membrane physiology, cell signalling, and cell adhesion, which phenotypically results in defects in cellular proliferation.

Furthermore, we show that svtRNA1-2–mediated NRS relies on both seed-plus-complementary-paired region hybridisation features and the presence of a 5-nt loop protrusion on target RNA. This study is the first report to describe the mechanism of svtRNA1-2 biogenesis and its regulatory role in NRS.

# Results

### vtRNA1-2 is processed by Dicer into small RNA fragments

In order to characterise fragments derived from vtRNA and to examine the potential role of Dicer in the processing of vtRNA, we performed RNA sequencing of small RNA (sRNA) isolated from HEK293T WT and Dicer knockdown (KD) (HEK293T cells with integrated inducible shRNA) (Fig S1A) cells (Di Fazio et al, 2022). A principal component analysis outlined clear differences between the two conditions and tight clustering of the replicates (Fig S1B). The size distribution of mapped reads showed a distinct Dicer-dependent peak between 22 and 25-nt-long fragments, including reads corresponding to mature miRNA (21–23 nt) (Fig S1C). The small difference in this peak between WT and Dicer KD is most likely due to the presence of the induced shRNA in the Dicer KD sample (used to induce Dicer KD), which might mask the absence of a proportion of mature miRNA. Among the identified sRNAs, the most abundant species were miRNA (making up 47% of the population), followed by tRNA (23%). We also identified the four species of vault RNA (Fig 1A and Table S1), with >22-nt-long fragments. Interestingly, fragments derived from vtRNA1-2 were mostly 22–24 nt long, suggesting that they might be Dicer products (Fig S1D). Indeed, moving average (MA) analysis showed that fragments derived from vtRNA1-2 and vtRNA2-1 were significantly down-regulated in Dicer KD (*P*-adj < 0.05), suggesting Dicer is important for their biogenesis (Figs 1B and S1E). Interestingly, we identified more svtRNA1-1 reads upon Dicer KD when compared to WT, whereas the expression of vtRNA1-3 was not significantly altered between these samples (Fig 1B), suggesting

that Dicer is only involved in the processing of vtRNA1-2 and vtRNA2-1. These data are in agreement with additional analysis of small RNA-seq from a previously published dataset in WT and Dicer KD conditions (GSE55333) (Rybak-Wolf et al, 2014) (Fig S1F and Table S2).

Next, we calculated the per-base read coverage of differentially expressed vtRNA in both WT and Dicer KD conditions. We identified regions along the vtRNA gene sequence with high read coverage, located at the 3′ end of vtRNA1-2 and the 5′ end of vtRNA2-1, which were decreased in Dicer KD, indicating that vtRNA1-2 and vtRNA2-1 can be processed into small vault RNA fragments in a Dicer-dependent manner (Figs 1C and S1F).

To validate our sequencing data experimentally, we purified recombinant Dicer protein from insect cells (Fig S2A and B) and performed in vitro Dicer cleavage assays on vtRNA1-1 and vtRNA1-2, using tRNA$^{Pro}$ as a positive control and tRNA$^{Ala}$ as a negative control (Di Fazio et al, 2022). Our data show that Dicer can indeed cleave vtRNA1-2 in vitro, although less efficiently than tRNA$^{Pro}$. We did not detect Dicer cleavage activity on tRNA$^{Ala}$ (negative control) and vtRNA1-1 (Fig 1D). It should be noted that vtRNA1-2 resembles more of tRNA-like conformation rather than of a pre-miRNA molecule with a defined stem–loop structure and 2-nt 3′ overhang, suggesting that vtRNA1-2 is a non-canonical Dicer substrate. Indeed, the presence of 4 Uridines at 3'-end and the tRNA-like stem-loop region of the vtRNA1-2 molecule might affect proper interaction with Dicer PAZ, dsRBD and helicase domains. Interestingly, the presence of a hammerhead-like stem loop structure in vtRNA1-1 might be a pivotal hindering factor for Dicer processing.

Next, we analysed the reactivity scores from Luo et al (2021) for each nucleotide on the vtRNA1-2 sequence and identified a cluster of highly reactive nucleotides between nucleotides 20 and 50, suggesting that this might be the region of vtRNA1-2 interaction with processing factors (Fig S2C). The newly identified svtRNA1-2 fragment was visualised on the vtRNA predicted secondary and tertiary structures (Figs 1E and S2C). Interestingly, detected reactive nucleotides on vtRNA1-1 were separated into two smaller clusters (around nucleotide 40 and nucleotide 60), which might suggest a difference in the processing of vtRNA1-1 and vtRNA1-2.

All four vtRNA sequences are highly similar in the 5′ and 3′ regions but vary in the central domain (between nucleotides 30 and 60), which could be relevant to the differences in reactive nucleotide distribution (Fig S3A). Interestingly, when we analysed vault RNA1-2 sequence conservation, we confirmed that the 5′ and 3′ regions of vtRNA are highly conserved across vertebrates, whereas the middle part is not (Fig S3B). In mice, vtRNA with unknown function is encoded by only one gene, VAULTRC 5, which is longer (143 nt) than any of the human vtRNA. Because of a high conservation score, we hypothesised that the processing of vtRNA into svtRNA might be a general evolutionarily conserved mechanism. To verify this hypothesis, we analysed a collection of small RNA-seq data from mice (spleen, liver, lungs, muscle, brain, and pancreas) (GSE119661). Interestingly, we identified reads localising to a specific region at the 3′ end on the template vtRNA, revealing novel small vtRNA fragments (Fig S3C). Furthermore, we noticed the different expression patterns throughout the analysed tissues, which resemble the tissue-specific expression of vtRNA in humans. These results show the presence of vtRNA-derived fragments in

mouse tissues, supporting the evidence that svtRNAs are not degradation products but instead specific biologically functional species. Altogether, these data confirm the existence of an evolutionarily conserved novel Dicer-dependent svtRNA1-2.

## svtRNA1-2 associates with Argonaute proteins and is localised predominantly in the nucleus

Next, we set out to test whether svtRNA1-2 is functional. We have previously identified that a group of tsRNA can mediate NRS (Di Fazio et al, 2022). As vtRNA shares some similarities to tRNA, such as length, Nsun2-dependent $m^5C$ modification, Dicer processing, and RNAPIII transcription, we assessed the potential role of svtRNA1-2 in gene silencing. First, we investigated whether svtRNA1-2 fragments associate with Argonaute proteins by analysing publicly available datasets of Ago1-4 (GSE21918) PAR-CLIP (photoactivatable ribonucleoside-enhanced crosslinking and immunoprecipitation) and of Ago2/3 RIP-seq (GSE55333) (Table S2). Interestingly, we detected significant enrichment of Dicer-dependent svtRNA1-2 in the Ago2 PAR-CLIP. Fragments derived from vtRNA1-1 were present in the Ago2 PAR-CLIP data only at low levels, and svtRNA1-3 and svtRNA2-1 were not detected. We used microRNA-21 (miR-21) as a positive control (Figs 2A and S4A). We also observed svtRNA1-2 in the other Argonaute PAR-CLIP data (Ago1, Ago3, and Ago4) and in the Dicer PAR-CLIP dataset (Table S2 and Fig S4A).

It has been shown that Ago2, the only member of the Argonaute family with well-characterised catalytic activity (Ming et al, 2007), can carry out its function in both the nucleus and the cytoplasm (Kalantari et al, 2016; Sarshad et al, 2018; Gong et al, 2021). Given that we have shown that tsRNAs function in the nucleus (Di Fazio et al, 2022), we tested whether svtRNA1-2 might be present in a nuclear functional complex by analysing RNA bound to cytoplasmic and nuclear Ago2 in MCF-7 cells (GSE66665) (Table S2). Surprisingly, the svtRNA1-2 fragment was enriched in both nuclear and cytoplasmic Ago2 fractions. We used miR-21 (Kriegel et al, 2018) as a nuclear positive control (Fig S4B).

Next, we wished to visualise endogenous vtRNA1-2 and svtRNA1-2 using FISH. We designed probe 1 complementary to the middle region of vtRNA1-2 (the most unique region for vtRNAs in terms of sequence) for detection of full-length vtRNA1-2. We also designed probe 2 complementary to the 3' region of vtRNA1-2 for detection of both full-length vtRNA and svtRNA1-2 (Fig 2B). First, we tested whether these probes specifically recognise vtRNA1-2 and not vtRNA1-1. Incubation of the probes with in vitro transcribed vtRNA1-2 and vtRNA1-1, followed by Northern blot, shows that both probes are specific for vtRNA1-2 and do not hybridise to vtRNA1-1 (Fig 2B). Next, we fluorescently labelled both probes and performed the FISH experiment in HEK293T WT and vtRNA KO cells. We observed that probe 1 gave a predominantly cytoplasmic signal, corresponding to full-length vtRNA1-2, whereas in contrast, the signal for probe 2, corresponding to vtRNA1-2 and svtRNA1-2, was more nuclear. This was the case for FISH with incubation with single probes (1 or 2) or with double probes (1 + 2). The quantification of FISH signals confirms that the probe 2 signal is significantly more nuclear, whereas the probe 1 signal is more cytoplasmic. Using probes 1 and 2 in vtRNA KO cells did not result in positive signals, further confirming the specificity of the FISH experiment (Fig 2C and D). Closer

inspection of the FISH signals also revealed the formation of probe 2 clusters, which were not visible with probe 1 (Fig S5A). These data suggest that endogenous vtRNA1-2 resides more in the cytoplasm, whereas svtRNA1-2 is localised predominantly in nuclear clusters. Next, we transfected HEK293T cells with fluorescently labelled single- and double-stranded svtRNA1-2 and followed its cellular localisation at 4, 6, and 24 h post-transfection. We observed mostly nuclear localisation of single-stranded and double-stranded svtRNA1-2 at all time points tested (Fig S5B and C). We noted that transfected svtRNA was forming clusters, similar to transfected tsRNA (Di Fazio et al, 2022), but not siRNA. These foci could potentially suggest functional aggregates and possibly are an exaggeration of the clusters formed by endogenous svtRNA1-2, as the synthetic svtRNA1-2 is added to the cells in excess.

Taken together, these data show that svtRNA1-2 fragments bind to Ago2 in both cellular compartments and can be detected in the nucleus, suggesting that it might play a role in nuclear gene silencing.

## svtRNA1-2 targets selected genes at the nascent RNA level

The Dicer dependency and association of svtRNA1-2 with Ago2, along with its nuclear localisation, prompted us to investigate whether svtRNA1-2 could regulate gene expression on a nascent RNA level, similar to tsRNA. We employed ChrRNA-seq to characterise the nascent transcriptome of HEK293T cells in WT, Ago2 KD, and Dicer KD conditions (Di Fazio et al, 2022) (Fig 3A and Table S2). Differential gene expression analysis revealed that Ago2 and Dicer KD had a broad effect on the transcriptome, with 1,346 genes down-regulated and 2,568 up-regulated in Ago2 KD, and 1,052 genes down-regulated and 2,391 up-regulated in Dicer KD ($log_2FC > 1/log_2FC < -1$ and $P$-adj < 0.001). The up-regulated genes in Ago2 and Dicer KD conditions included mostly protein-coding genes and long non-coding RNA (Fig 3B and C and Table S3). In order to identify the role of svtRNA1-2 in gene silencing, we analysed the intersection of up-regulated protein-coding genes in both Ago2 and Dicer KD conditions (1,272 genes), as we showed that svtRNA1-2 is Dicer-dependent and binds to Ago2 (Fig 3D). We employed the well-known and widely used miRanda software (Enright et al, 2003) for in silico prediction of svtRNA1-2 gene targets through sequence complementarity, similar to miRNA binding sites, with restricted seed length and binding energy (see the Materials and Methods section). Interestingly, we found that the highest number of predicted gene targets is unique for svtRNA1-2 (227 genes; Table S3), when compared to svtRNA1-1 (78 genes) or svtRNA1-3 (83 genes) (Fig 3D). We used microRNA-4487 (miR-5587) as a negative control, as it has a sequence similar to svtRNA (Figs 3D and S4C). Gene ontology enrichment analysis of the predicted svtRNA1-2 target genes revealed pathways involved in cell migration, motility, and regulation of proliferation and growth (Fig 3E). Indeed, the transmembrane receptor tyrosine-protein kinase ERBB4 was among the top 5 predicted target genes with the highest $log_2$ fold change (Table S3). To investigate this further, we mapped the predicted binding sites along the transcript lengths and found a distinct density pattern for svtRNA1-2 distributed along the body of the genes. In contrast, the other svtRNA fragments were mapped mostly to the beginning of the transcripts, resembling miRNA-mediated

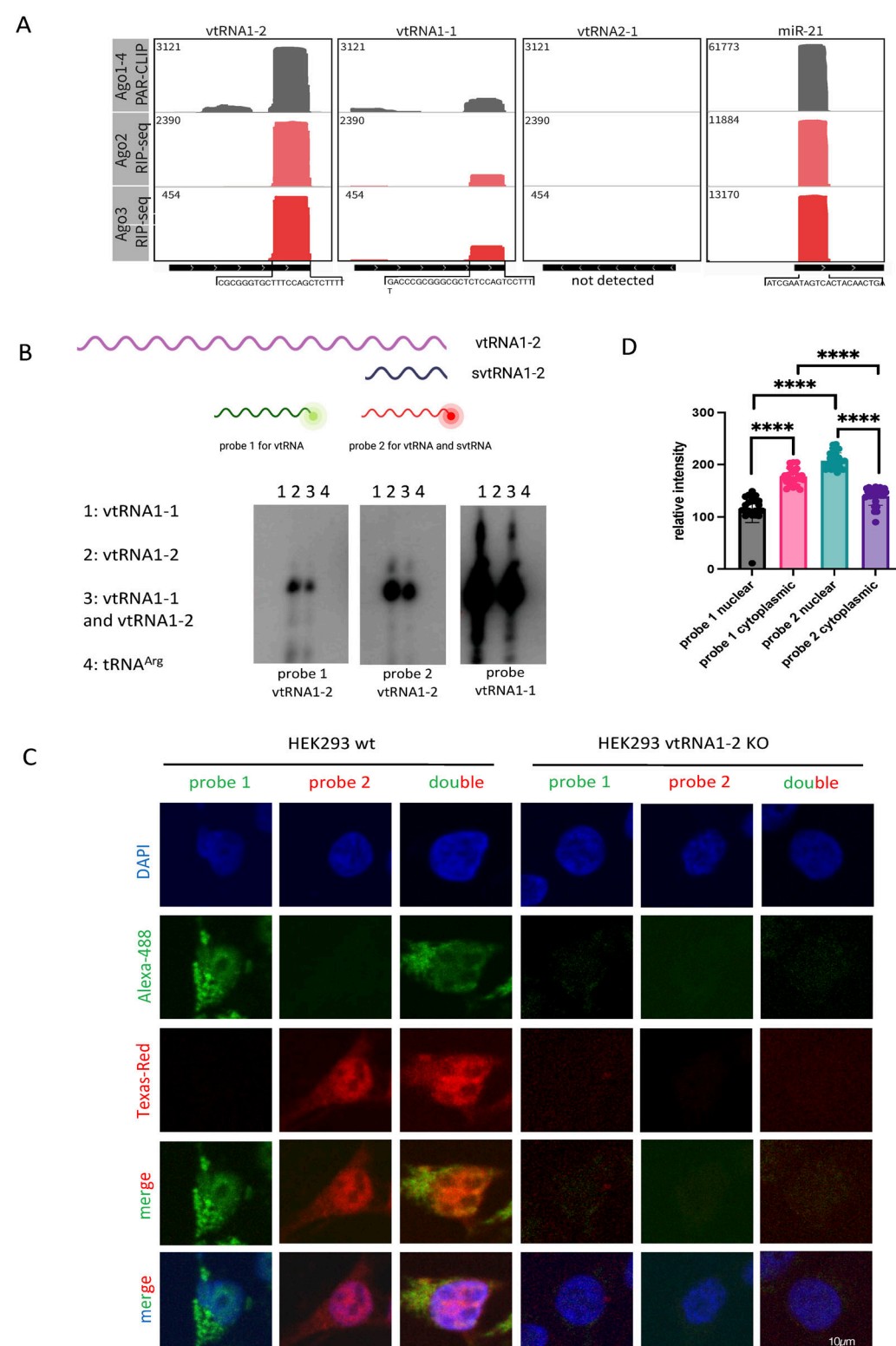

**Figure 2. svtRNA1-2 is bound to Argonaute proteins.**
**(A)** Integrative Genomics Viewer tracks of AGO1-4 PAR-CLIP and AGO2/3 RIP-seq show the association of AGO proteins with svtRNA1-2 (data values are shown in the top left corner). miR-21 was used as a positive control. The bottom black line indicates the length of the transcript, and the sequence below represents the interacting fragment. Note: no reads were detected for vtRNA2-1. **(B)** Diagram illustrating the position of probes 1 and 2 with respect to vtRNA1-2 and svtRNA1-2. Representative Northern blot showing signals for vtRNA1-1 and vtRNA1-2 when incubated with probes 1 and 2 designed for vtRNA1-2. **(C)** Representative FISH confocal images showing

gene activation. It would be interesting to test whether vtRNA1-1 and vtRNA1-3 could lead to gene activation (Fig 3F). These data suggest different roles of svtRNA1-2 and svtRNA1-1/1-3. Furthermore, we observed that svtRNA1-2 mapped to intronic regions with the highest occurrence, similar to tsRNA, suggesting a potential role of svtRNA1-2 in NRS (Fig 3G). In summary, we identified genes that are up-regulated upon Dicer and AGO2 KD on a nascent level and are in silico predicted to be targeted by svtRNA1-2 mostly within introns.

## svtRNA1-2 knockout leads to defects in cellular proliferation

To investigate the molecular function of svtRNA1-2 further, we employed an RNA-guided CRISPR/Cas12a system targeting vtRNA1-2 with single guide RNAs (sgRNAs) in HEK293T cells (Fig S6A and B). We successfully generated individual vtRNA1-2 KO clones and confirmed partial deletion of vtRNA1-2 by sequencing and PCR (KO results in a shorter genomic fragment). As vault loci show relatively high sequence similarity, we tested whether the other vtRNA loci remain intact in vtRNA1-2 KO. Indeed, vtRNA1-1, vtRNA1-3, and vtRNA2-1 genomic loci were not affected by vtRNA1-2 deletion (Fig 4A). Next, we wished to validate the svtRNA1-2 predicted targets. We performed a RT–qPCR using RNA isolated from WT and stable vtRNA1-2 KO cell lines to validate the expression levels of ERBB4, ADAM12, KCNH1, and PLXNA4 (unique targets for svtRNA1-2), and PARP12 (a unique target for svtRNA1-1, used as a negative control). We show that the expression of all svtRNA1-2 predicted targets was significantly increased at both the nascent and steady-state RNA levels in vtRNA1-2 KO cells, whereas the expression of PARP12 remained unaffected (Fig 4B and C).

Interestingly, we observed impaired cellular proliferation in vtRNA1-2 KO cells, when compared to WT cells, as shown by the clonogenic survival assay and the MTS proliferation assay (Fig 4D and E). The decreased cell viability and reduced proliferation of vtRNA1-2 KO cells suggest that vtRNA1-2 plays an important role in normal cellular function. To validate and investigate this phenotype further, we performed a rescue experiment with plasmids expressing vtRNA1-2 and svtRNA1-2, and luciferase plasmids (as a negative control). We show that a vtRNA1-2 plasmid (providing both full and small vtRNA1-2) leads to significant rescue of a proliferation phenotype. Furthermore, svtRNA1-2 alone also leads to partial but significant rescue, as well as vtRNA1-2 alone (vtRNA1-2 in Dicer KD in vtRNA1-2 KO cells). These data suggest that both vtRNA1-2 and svtRNA1-2 are relevant to a vtRNA1-2 KO proliferation phenotype (Fig 4F).

## Deletion of vtRNA1-2 alters the expression of genes associated with cellular membrane function

To further explore the role of svtRNA1-2 in nascent RNA silencing, we isolated chromatin-associated RNA from HEK293T WT and vtRNA1-2 KO cells and performed ChrRNA-seq (Fig 5A). The purity of the fractions was confirmed by Western blotting using H3 as a chromatin marker and tubulin as a cytoplasmic marker (Fig

S7A). A principal component analysis outlined clear differences between the two conditions and tight clustering of the replicates (Fig S7B and C). Differential expression analysis revealed 547 down-regulated and 944 up-regulated genes (mostly protein-coding genes) in vtRNA1-2 KO cells (Figs 5B and S7D and E). Closer inspection showed that ERBB4 was among the up-regulated genes in vtRNA1-2 KO, providing further validation for svtRNA1-2–mediated down-regulation of ERBB4 (Fig 5B and C).

Gene ontology enrichment analysis of genes up-regulated in vtRNA1-2 KO cells revealed an association of protein-coding genes with biological processes such as regulation of membrane transport and potential (Figs 5D and S8A), whereas down-regulated genes were not enriched in these categories (Fig S8B). Interestingly, enrichment analysis that includes both up- and down-regulated genes revealed a significant association of differentially expressed genes upon vtRNA1-2 KO with early and late oestrogen response, potentially, at least partially, because of deregulated ERBB4 (Fig 5E). Finally, the KEGG pathway analysis confirmed an altered ERBB4 signalling pathway (Fig 5F).

These data suggest that vtRNA1-2 is involved, either directly and/or indirectly, in the regulation of membrane proteins through the silencing function of svtRNA1-2.

## svtRNA1-2 target genes are associated with cellular membrane function

Next, we used miRanda software to predict target genes for svtRNA fragments in vtRNA1-2 KO up-regulated protein-coding genes. As expected, we identified most of the unique target genes for svtRNA1-2 (Fig S9A). Closer inspection of miRanda predictions revealed 139 target genes and 205 target sites for svtRNA1-2 (Fig S9B). Comparing miRanda with another target site prediction software PITA, we observed a good overlap of unique target genes; however, more stringent miRanda settings resulted in more predicted genes (Fig S9C and Tables S4 and S5). miRanda software is based on a dynamic programming local alignment between the query sequence and the reference sequence. High-scoring alignments estimate the thermodynamic stability of RNA duplexes based on these alignments. The PITA tool assesses the accessibility of the target site. It evaluates the free energy gained from miRNA–mRNA pair formation and the energy cost of making the target accessible to the miRNA and computes the difference between these two parameters. Therefore, for further analysis, we used targets identified by miRanda. A metagene alignment showed even distribution of svtRNA1-2 target sites, along the gene length, whereas svtRNA1-1 and svtRNA1-3 target sites were accumulated at the beginning of the genes, similar to what we observed in Fig 3F (Fig S9D). Finally, svtRNA1-2 target sites were predominantly mapped to the intronic regions. These results are consistent with the previous predictions (Fig S9E).

fluorescent signals for probes 1 and 2 in HEK293 WT and vtRNA1-2 KO cells. DAPI was used to stain nuclei. **(D)** Bar chart showing relative fluorescence intensity levels of probes 1 and 2 in the nucleus and the cytoplasm. Error bar = mean ± SEM; significance was determined using a multiple unpaired $t$ test, ****$P \le 0.0001$.

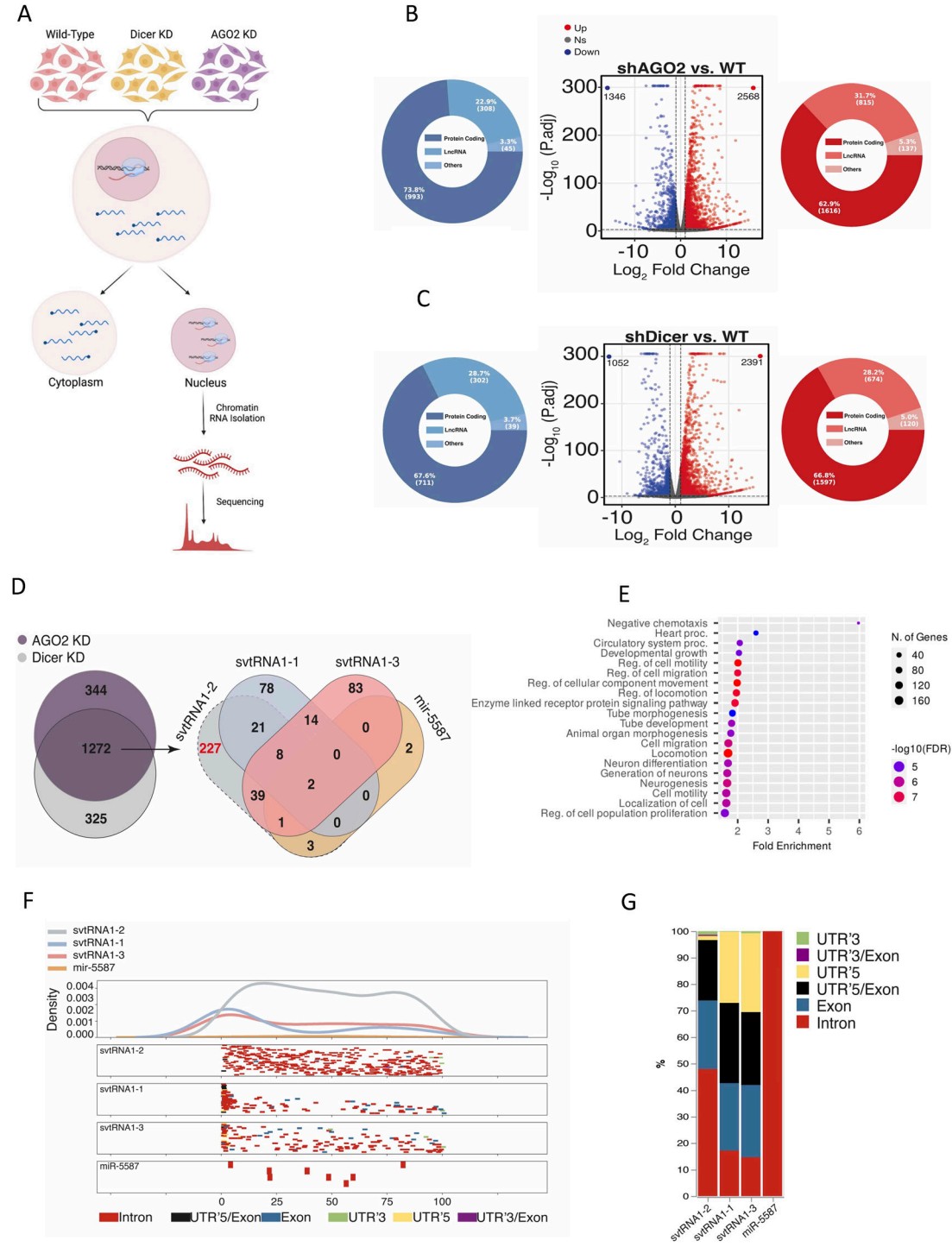

**Figure 3. svtRNA1-2 targets selected genes at the nascent RNA level.**
**(A)** Schematic representation of the ChrRNA-seq method. **(B)** Volcano plots of differentially expressed genes in AGO2 KD conditions. The red points and blue points show up-regulated genes and down-regulated genes with log₂FC > 1 and log₂FC < −1 and P-adj < 0.001, respectively. Pie charts show different gene types of up (red)- or down-(blue)-regulated genes. **(B, C)** As in (B) using Dicer KD samples. **(D)** Venn diagram shows an overlap of up-regulated protein-coding genes from Dicer and AGO2 KD conditions (left). Venn diagram shows the number of predicted target genes for svtRNA1-1/2/3 and miR-5587 (right). svtRNA1-1/3 and miR-5587 were used as controls as their sequence similarity has the highest score with svtRNA1-2. **(E)** GO analysis of vtRNA1-2 predicted targets up-regulated in AGO2 and Dicer KD samples. **(F)** Metagene distribution of predicted svtRNA binding sites mapped along normalised target transcript length and their corresponding gene features. **(G)** svtRNA binding site counts normalised to the length of a feature displayed in overall percentage.

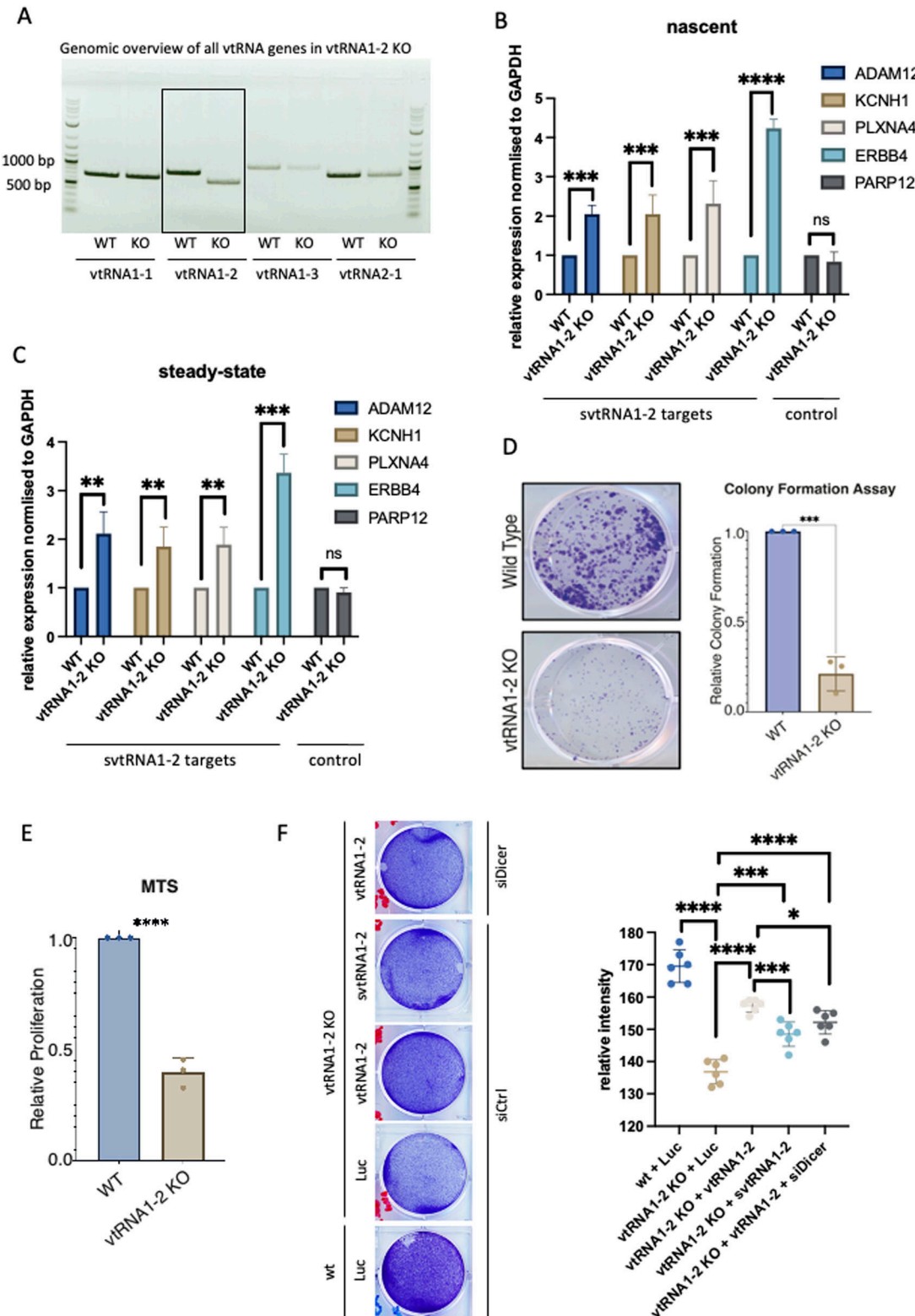

**Figure 4. vtRNA1-2 Knockout leads to a severe cellular phenotype.**
**(A, B)** PCR analysis from purified genomic DNA from WT and vtRNA1-2 KO showing each vault RNA. (B) qRT–PCR quantification of nascent RNA levels of ADAM12, KCNH1, PLXNA4, and ERBB4 as predicted targets of svtRNA1-2, and PARP12 as a predicted target of svtRNA1-1 (used as a negative control) in WT and svtRNA1-2 KO cells normalised to GAPDH. The data represent the mean fold changes from three independent experiments (****$P$ < 0.0001; ***$P$ < 0.001; and n.s, not significant). **(B, C)** As in (B), showing steady-state RNA levels. **(D)** Clonogenic assay of WT and vtRNA1-2 KO cells. The cells were stained and counted after 10 d of growing (***$P$ < 0.001). **(E)** MTS assay showing

Focusing on unique svtRNA1-2 target genes, we analysed whether previously predicted svtRNA1-2 target genes that were up-regulated upon Dicer and Ago2 KD and predicted to be targeted by svtRNA1-2 were also up-regulated in vtRNA1-2 KO cells. We found 29 common genes including ERBB4 (Figs 6A and S9F and Table S6). To analyse whether this is statistically significant, we performed further analysis as follows: first, we used the 584 up-regulated protein-coding genes in the vtRNA1-2 KO sample, ran a random subset of protein-coding genes 1,000 times, and overlapped this random subset with the 584 up-regulated genes. This resulted in Min.: 0,000; 1st Qu.: 7,000; Median: 9,000; Mean: 8,936; 3rd Qu.:11,000; and Max. :19,000, suggesting that the intersection of 29 up-regulated vtRNA1-2 KO genes with the 227 up-regulated genes in Dicer/Ago2 KD samples is indeed statistically significant. Finally, overall probability statistics revealed the following: Set1: 227; Set2: 584; overlap: 29; total number of genes: 14,759; representation factor: 3.2; and $P < 2.510 \times 10^{-8}$. It should be noted that a representation factor >1 indicates more overlap than expected of two independent groups, a representation factor <1 indicates less overlap than expected, and a representation factor of 1 indicates that the two groups overlap by the number of genes expected for independent groups of genes.

Interestingly, proteins encoded by these genes were mostly localised in the cellular membrane (Fig 6B). Indeed, STRING (Szklarczyk et al, 2019) analysis identified 14 of 29 to be involved in signalling and 17 of 29 to be glycoproteins (Fig 6C). Glycoproteins are membrane proteins that play a key role in cell migration, proliferation, and adhesion (Gu et al, 2012). Indeed, a complex pathway analysis revealed that svtRNA1-2 target genes are involved in cell adhesion regulation (Fig 6D). To validate the result, we seeded an equal number of WT and vtRNA1-2 KO cells and monitored their adhesion abilities. We observed elongated vtRNA1-2 KO cells with reduced cell-to-cell contact, most likely as a potential consequence of negatively regulated cell-to-cell adhesion (Fig 6E).

Overall, these data demonstrate that vtRNA1-2 is involved in the regulation of nascent RNA of protein-coding genes that are associated with cell membrane physiology.

### svtRNA1-2 targets intronic regions through a seed-plus-complementary-paired region and target loop protrusion architecture

Analysis of the miRanda output revealed that svtRNA1-2 hybridises to the predicted targets in a miRNA-like fashion. Thermodynamic asymmetry between the two ends of the svtRNA1-2–target duplex, results in near-perfect complementarity at the 5′ seed region, and the 3′ supplementary base pairing features are depicted in the predicted targets (Table S4). Interestingly, the analysis of svtRNA1-2–target hybridisation features also revealed the following: (1) the nearly perfect base pairing extends over the 5′ seed region (2–8) up to a nucleotide 18 with wobble base pairs and (2) there is the presence of a 5-nt loop in the target RNA for 179 of 205 predicted targets. Moreover, the 5-nt loop shows a conserved sequence (CAUCA/CAUUA) within the 179 predicted target genes (Table S4).

To determine the relationship between the svtRNA1-2 sequence and target hybridisation, a comparative analysis of svtRNA1-2 WT and scrambled svtRNA1-2 sequences (5′ shuffled, 3′ shuffled, and swapped) was performed on the group of the up-regulated genes in vtRNA1-2 KO (Fig7A; 5′ seed sequence is highlighted in orange, 3′ complementary sequence is highlighted in blue, and shuffled bases are depicted in red; svtRNA1-1 was used as a negative control). To prevent a biasing artefact, we kept the base composition of shuffled svtRNA sequences similar (Fig S10A). All svtRNA sequences were run through miRanda software against up-regulated genes identified in vtRNA1-2 KO to obtain their target predictions. Mapping of 5′ shuffled svtRNA1-2 sequence resulted in a complete loss of targets when compared to the svtRNA1-2 WT sequence, whereas 3′ shuffled svtRNA1-2 mapped to only three targets.

The swapped svtRNA1-2 mapped to 158 targets, which partially overlap with the svtRNA1-2 WT predicted targets (39 of 138) (Fig 7B). However, unlike the svtRNA1-2 wt, the swapped svtRNA1-2 sequence predominately hybridised to the exons of the target genes (Figs 7C and S10B), indicating that the predicted swapped svtRNA1-2 targets have different binding sites along the nascent RNA target when compared to svtRNA1-2 WT sequence. Of note, the 5-nt loop structure was not detected within the group of targets predicted for the swapped svtRNA1-2 (Fig 7D and E). Most (147 of 236) of the predicted swapped svtRNA1-2 target sites showed a length of 23 or 24 nucleotides that corresponds to the absence of the loop structure (Fig 7E).

To gain further insight into the molecular mechanism of svtRNA1-2–mediated NRS, several modified svtRNA1-2 molecules were transfected into the human breast cancer cell line MCF-7, and the expression level of nascent ERBB4 RNA was assessed by RT–qPCR, 24 h post-transfection (Fig 7F).

Specifically, we employed the single-stranded svtRNA1-2 molecules: 5′ scrambled at nucleotides 2–8, 2–12, and 2–16; 3′ scrambled at nucleotides 12–18; swapped at 5′/3′ ends; and svtRNA1-2 bearing perfect complementarity with the target gene (svtRNA1-2 with no loop). Interestingly, the nascent RNA level of ERBB4 was significantly reduced by svtRNA1-2 WT transfection, but was not altered in the presence of svtRNA1-2 with no loop (perfectly complementary), indicating that the 5-nt loop on the target RNA is pivotal for NRS of ERBB4. Moreover, loss of silencing upon transfection of 5′ and 3′ scrambled svtRNA1-2 sequences indicates that complementary of both the 5′ and 3′ regions is fundamental for effective NRS. The absence of silencing with the swapped svtRNA1-2 also suggests that silencing of ERBB4 is sequence- and site-specific. Finally, Bcl-2 nascent RNA levels,

proliferation of WT and vtRNA1-2 KO cells after 7 d (****P < 0.0001). **(F)** Left, representative image of proliferation assay of WT and vtRNA1-2 KO cells, transfected with plasmids expressing either control luciferase (Luc), full vtRNA1-2, or svtRNA1-2 in the presence of control siRNA (siCtrl) or siRNA targeting Dicer (siDicer). Right, quantification of the relative intensity of a signal in the rescue proliferation assay (****P < 0.0001; ***P < 0.001; and *P < 0.005).

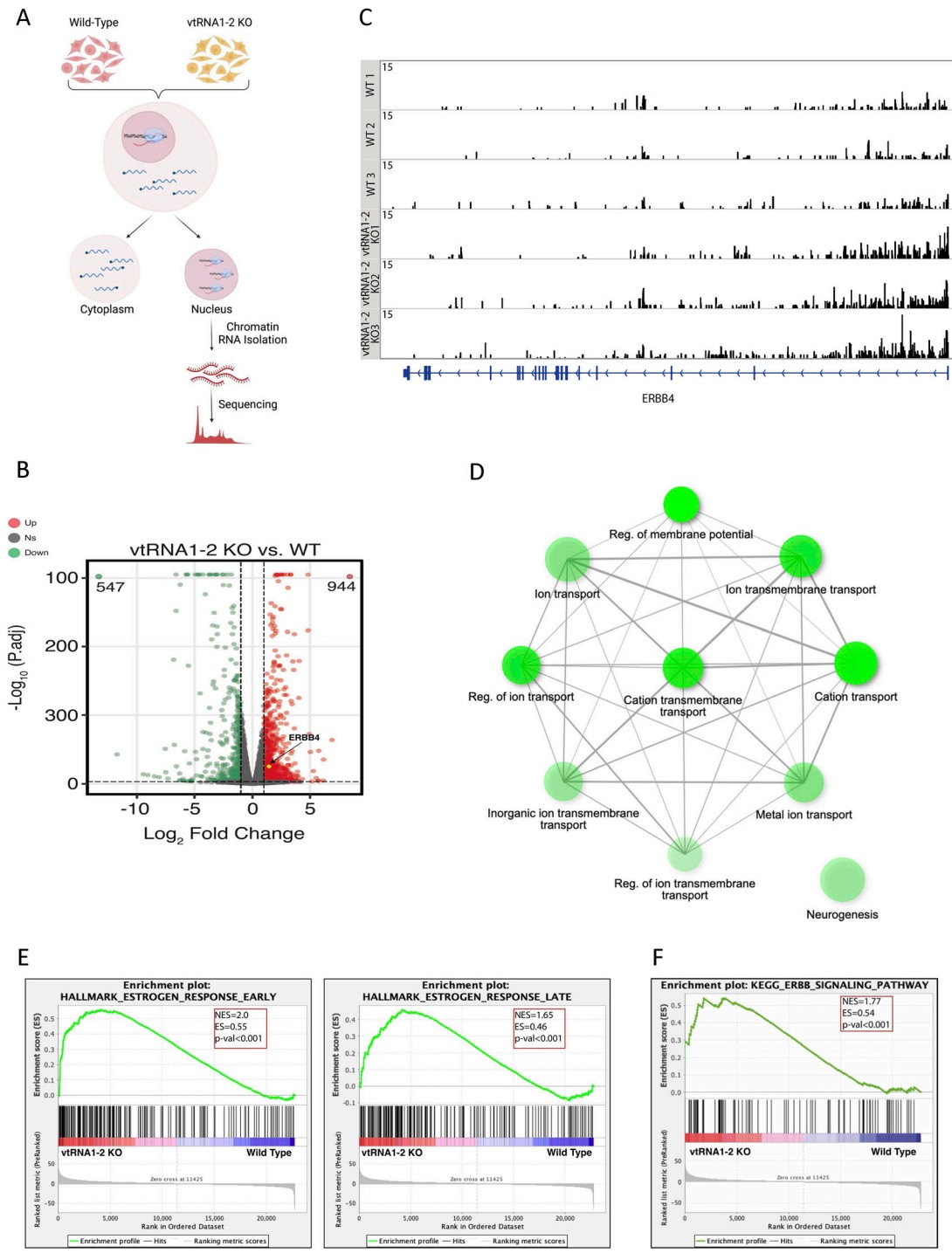

**Figure 5. Deletion of vtRNA1-2 alters the expression of genes associated with cellular membrane function.**
**(A)** Schematic representation of the ChrRNA-seq method. **(B)** IGV tracks of ChrRNA-seq from WT and vtRNA1-2 KO cells displaying the ERBB4 reads. The bottom blue line indicates the composition of the ERBB4 gene. **(C)** Volcano plots of differentially expressed genes in vtRNA1-2 KO and WT conditions. The red points and green points show up-regulated genes and down-regulated genes with $\log_2$FC > 1 and $\log_2$FC < −1 and $P$-adj < 0.001, respectively. The yellow point shows ERBB4. **(D)** GO analysis of up-regulated genes in vtRNA1-2 KO when compared to WT. **(E)** Gene set enrichment analysis with a gene set from vtRNA1-2 KO cells using Hallmark gene sets in MSigDB. Genes are ranked by Wald's statistic values in a ChrRNA-seq experiment (KO versus WT). A positive enrichment score indicates higher expression after vtRNA1-2 KO. **(F)** Gene set enrichment analysis with a gene set from vtRNA1-2 KO cells using KEGG pathways.

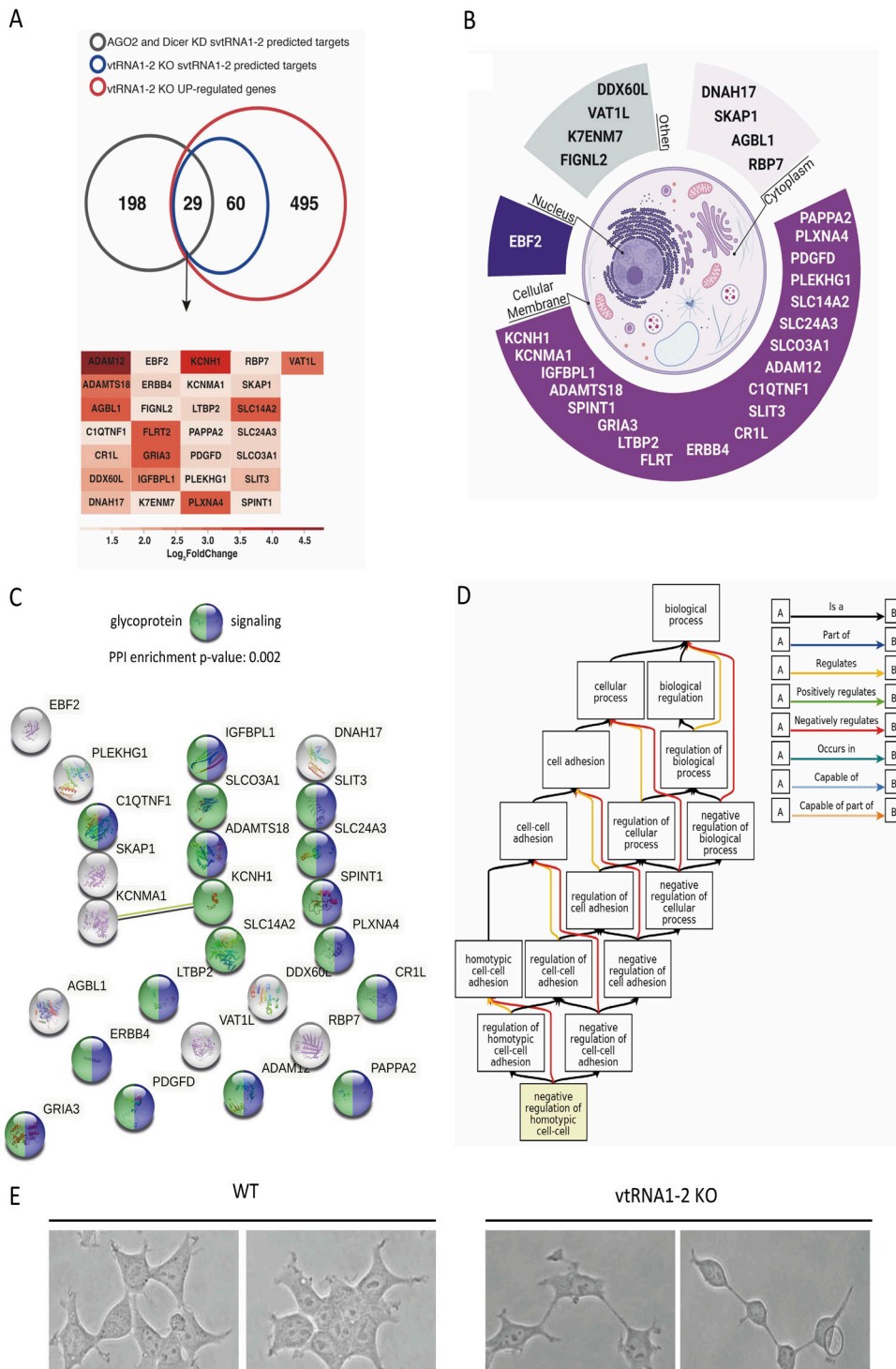

**Figure 6. svtRNA1-2 target genes are associated with regulation of cell adhesion.**
**(A)** Venn diagram showing an overlap between up-regulated protein-coding genes from vtRNA1-2 KO (compared with WT), vtRNA1-2 KO svtRNA1-2 predicted targets, and svtRNA1-2 predicted targets from up-regulated genes in AGO2 and Dicer KD cells (top). Tables showing names of 29 svtRNA1-2 target genes. Heatmap is showing fold of expression change for each target gene in vtRNA1-2 KO cells. **(B)** Pie chart showing cellular localisation of 29 svtRNA1-2 target genes. **(C)** STRING analysis showing interactions among 29 svtRNA1-2 target genes. Proteins involved in signalling are marked in blue, and glycoproteins are marked in green. **(D)** Pathway analysis showing an association of svtRNA1-2 target genes with negative regulation of cell adhesion. **(E)** Representative examples of bright-field microscopy showing differences in a cell adhesion phenotype in WT and vtRNA1-2 KO cells.

used as a negative control, were not affected by svtRNA1-2 molecules.

These data demonstrate that svtRNA1-2 possesses unique features that are required for specific intronic targeting for NRS.

## Discussion

In this work, we have identified Dicer-dependent small RNA fragments derived from vtRNA1-2 and vtRNA2-1. We also found vtRNA1-1 to be independent of Dicer processing in HEK293T cells, which is in

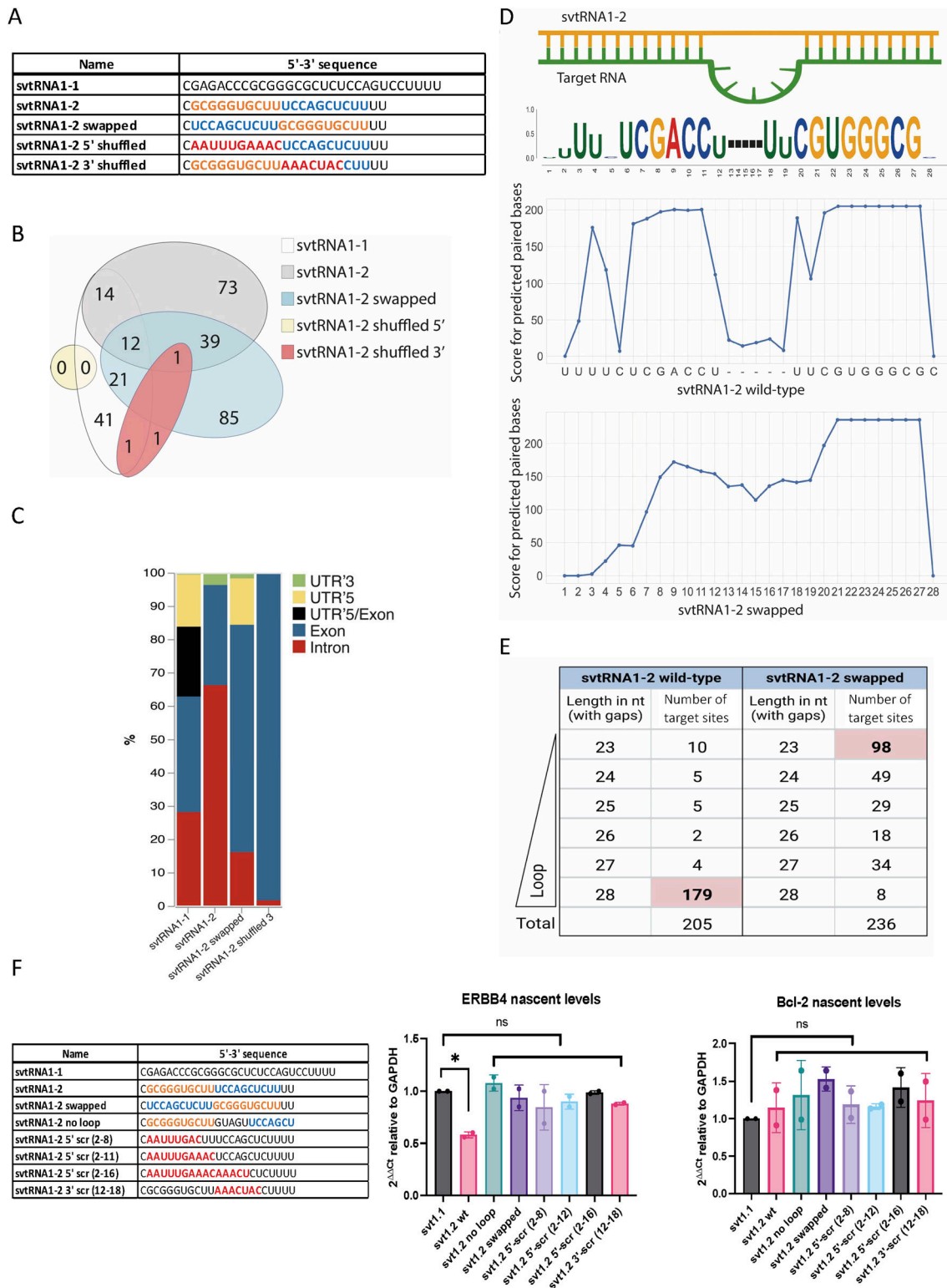

**Figure 7. svtRNA1-2 targets intronic regions through a seed-plus-complementary-paired region and target loop protrusion architecture.**
**(A)** Sequence representation of svtRNA1-1, svtRNA1-2 WT, svtRNA1-2 swapped, svtRNA1-2 shuffled 5′, and svtRNA1-2 shuffled 3′. **(B)** Venn diagram showing an overlap of target genes predicted for each sequence. **(C)** Stacked bar plot showing predicted svtRNA binding sites mapped to different gene features with normalised counts to the length of a feature. Displayed in overall percentage. **(D)** Sequence logo of significantly enriched bases in svtRNA1-2 predicted target sites (top). Line plot representing scores for predicted paired bases of svtRNA1-2 to their target sites (middle). Line plot representing scores for predicted paired bases of swapped svtRNA1-2 to their target sites (bottom). **(E)** Table showing the correlation analysis of the loop/no loop and number of target sites. 23 nt represents an alignment with no loop, and 28 nt

line with previous observations by Rybak-Wolf et al (2014). However, it is worth noting that there are also contrasting results that show vtRNA1-1 as a Dicer substrate (Persson et al, 2009; Langenberger et al, 2013). This discrepancy could be potentially explained by the different sensitivity of the experimental techniques and also perhaps by differing efficiencies of Dicer KD. Furthermore, vtRNAs are also differentially expressed in various tissues (Stadler et al, 2009; Fort & Duhagon, 2021; Hahne et al, 2021; Gallo et al, 2022), so vtRNA1-1 might be processed by Dicer in other cell lines. The analysis of RNA bound to Argonaute proteins using two different experimental methods identified svtRNA fragments derived from vtRNA1-1/2 to be loaded onto Argonaute proteins, whereas svtRNA2-1 fragments were not detected. In contrast, recent work showed the binding of Ago2 to the 5′ vtRNA2-1 fragment by RT–qPCR in SH-SY5Y cells (Minones-Moyano et al, 2013). The potential explanation for this contrasting observation could be a differential expression of vault RNA in various tissues and pathological contexts. Genome-wide expression studies of sncRNA, including svtRNA, in different cell types or tissues, would allow the identification of more detailed vtRNA expression patterns and could provide important insights into their biogenesis.

We found svtRNA1-2 to be associated with Ago2 not only in the cytoplasm but also in the nucleus, its nuclear localisation was confirmed by confocal microscopy using FISH and synthetic fluorescently labelled svtRNA1-2. As vtRNA shares similar features with tRNA, we hypothesised that svtRNA1-2 could guide Ago2 in the nucleus to promote NRS (Di Fazio et al, 2022). Furthermore, Ago2 has previously been proposed to regulate gene expression in the nucleus (Sarshad et al, 2018).

To investigate the potential role of svtRNA1-2 in an NRS mechanism, we used ChrRNA-seq datasets from Dicer and Ago2 KD cells and predicted svtRNA1-2 binding sites. We mapped svtRNA1-2 with the highest occupancy to the intronic regions of many genes involved in cell membrane physiology, including ERBB4. Indeed, genomic deletion of vtRNA1-2 resulted in inhibition of cell growth and proliferation. Such dramatic defects in cell growth and proliferation were not previously seen in vtRNA1-1 and vtRNA1-3 KO cells (Bracher et al, 2020), suggesting that vtRNA1-2 might play a role in different cellular processes. Analysis of nascent RNA isolated from HEK293T WT and vtRNA1-2 KO further confirms the role of svtRNA1-2 in NRS.

Well-established rules for functional miRNA–target pairing classify target sites into three main categories: canonical sites that pair at both 5′ and 3′ ends, seed sites (depending on minimal 5′ end base pairing), and 3′ compensatory sites characterised by weak 5′ base pairing and strong compensatory pairing to the 3′ end (Alagia & Eritja, 2016). To assess the NRS requirements of the svtRNA1-2 molecule, we analysed svtRNA1-2 WT pairing conditions between svtRNA1-2 and the predicted targets. A large set of predicted targets for svtRNA1-2 WT showed perfect 5′ seed complementarity with extended wobble base pairs at the 3′ end. Nucleotide conservation analysis of the predicted targets (Fig 7D) highlighted the presence

of a GC-enriched segment at the 5′ end that might have additional roles such as the 5′ terminal oligo-guanine (TOG motif) described as a characteristic feature of some 5′ tsRNA (Shi et al, 2019). We cannot exclude that svtRNA1-2 might form intermolecular RNA G-quadruplex structures and gain further functions in translational inhibition, protein recognition, or cellular localisation. In addition, svtRNA1-2 central (10-11-12) positions predominantly follow the base pair scheme wobble–WC–wobble. It should be noted that most of the predicted targets have an unpaired loop of five nucleotides opposite to positions 11 and 12 of the svtRNA1-2, with a common sequence 5′-ACUAC or AUUAC. It has been demonstrated that Ago2 can accommodate a central loop present on the target gene, without loss of binding affinity (Becker et al, 2019). Furthermore, we hypothesised that the target loop might be recognised as a consensus motif by some proteins. To further evaluate the minimal pairing requirements between svtRNA1-2 and the targets, we designed two svtRNA1-2 molecules with a shuffled sequence at either the 5′ end or the 3′ supplementary region (shuffled svtRNA1-2 5′ and shuffled svtRNA1-2 3′, respectively). Interestingly, the svtRNA1-2 shuffled 5′ designed to follow the rules of thermodynamic asymmetry, present in large numbers of miRNA molecules, did not have any common target genes with svtRNA1-2 WT (Fig 7B). Loss of the common targets could be connected to higher free energy values of shuffled svtRNA1-2 5′ with respect to the WT svtRNA1-2. Shuffled svtRNA1-2 at the 3′ end showed similar results to the svtRNA1-2 shuffled at the 5′ end, resulting in no common targets with svtRNA1-2 WT (Fig 7B). These data demonstrate that complementarity at both the 5′ and 3′ ends of the svtRNA1-2 is crucial for the identification of target genes. Because we reasoned that the complete loss of common targets between WT and 5′ and 3′ shuffled svtRNA1-2 might be partially due to the miRanda free energy cut-off, we designed a svtRNA1-2 molecule with a swapped 5′ and 3′ seed sequence. From the analysis of the target predictions, we observed some increase in the putative targets but a dramatic change in the target sequence complementarity localisation. The swapped svtRNA1-2 hybridises within the exons of the predicted genes, in contrast to the svtRNA1-2 WT, which hybridises within introns via a loop protrusion. In silico analysis together with data on the ERBB4 nascent level expression showed that svtRNA1-2 engages with a target through a specific hybridization signature and NRS svtRNA1-2-mediated is dependent on the formation of seed-loop-complementary pairing architecture in vivo.

In conclusion, in this work, we uncovered a Dicer-dependent small RNA fragment derived from vault RNA1-2. We show that svtRNA1-2 mediates nuclear gene silencing by targeting the introns of protein-coding genes that are associated with cell membrane function (Fig S10C). Deletion of vtRNA1-2 results in a deregulated proliferation phenotype, which underpins the biological significance of vtRNA1-2. Finally, we show that svtRNA1-2 employs 5′ and 3′ complementarity with the formation of a loop protrusion within its target sequences, which is required for

---

contains a 5-nt-long gap across the alignment, loop (the longest loop observed). **(F)** Left, table of sequences used for transient transfection of MCF-7 cells. Right, bar chart showing nascent RNA levels of ERBB4 (svtRNA1-2 target gene) and Bcl-2 (control gene) measured by RT–qPCR. Samples were isolated 24 h after transfection. The values were normalised to the GAPDH. The data represent the mean fold changes from three independent experiments (*$P$ < 0.05 and n.s, not significant).

intronic binding. Although we cannot yet fully appreciate how the loop protrusion might be recognised and used for nascent RNA degradation, we demonstrate a novel biological role of vtRNA1-2 and its potential for future RNA-based cancer therapies.

# Materials and Methods

### Small RNA analysis

FASTQ files were trimmed to remove the 3′ adapter sequence (GATCGGAAGAGCA CACGTCTGAACTCCAGTCACCGATGTATCTCGTATGC CGTCTTCTGCTTG) and filtered based on the length of the read (15–30) using Cutadapt (v.1.8.3) (Marcel, 2011). The following data pre-processing reads were then aligned with STAR aligner (v.2.5.3a) (Dobin et al, 2013) to the human genome (hg38) downloaded from the GENCODE (v.34) basic human genome annotation. Custom sncRNA annotation file was built by merging annotation of miRNA from miRBase (v.22.1) (Kozomara et al, 2019), tRNA from GENCODE (v.34), and snoRNA, snRNA, miscRNA, scaRNA, rRNA, and vault RNA from the GENCODE (v34) (hg38) basic genome annotation. Reads overlapping the custom annotation were quantified using featureCounts (v.1.6.2) (Liao et al, 2014). DEGs were identified using DESeq2 (Love et al, 2014) with an adjusted $P < 0.05$. The vault RNA fragments were identified using bedtools (v.2.29.2) (Quinlan & Hall, 2010) genomecov with the following settings: –dz and –scale, which calculates the read coverage along the genome.

### 6xHis-Dicer purification

Low-passage *Spodoptera frugiperda* (Sf9) cells were seeded at $0.5 \times 10^6$ cells/ml and grown in suspension at 27°C. On the next day, cells reached a density of $1.5–2.5 \times 10^6$ cells/ml and were infected with P1 baculovirus at MOI = 1, generated with the pFastBac1-hDicer (#89144; Addgene) and the Bac-to-Bac expression system (Gibco). All steps of purification were carried out at 4°C. Cells were harvested 72 h post-infection by spinning at 800*g* for 15 min and lysed with 6x volume of lysis buffer (20 mM Tris–HCl, pH 8, 500 mM NaCl, 10% glycerol, 0.5% Triton X, 1 mM TCEP, 1x Complete protease inhibitor, 1x PMSF, and 1x leupeptin) and micrococcal nuclease (NEB), and rotated on a wheel for 30 min. The lysate was then sonicated four times for 30 s on/off at 15 $\mu$m, span at 13,000*g* for 45 min, and then filtered with a 0.2-$\mu$m filter. 250 $\mu$l of Ni-NTA agarose (QIAGEN) for 50 ml of cell culture was equilibrated with wash buffer (20 mM Tris–HCl, pH 8, 500 mM NaCl, 5% glycerol, 1 mM TCEP, 1x Complete protease inhibitor, 1x PMSF, 1x leupeptin, and 5 mM imidazole). Imidazole was added to the lysate up to 5 mM and then incubated with pre-equilibrated Ni-NTA agarose on a tube roller for 1 h. Agarose–lysate mixture was spun down gently, and resin was let set in a gravity flow column (Bio-Rad) and washed twice with wash buffer. 6xHis-Dicer was then eluted with elution buffer with a gradient of imidazole (20 mM Tris–HCl, pH 8, 500 mM NaCl, 5% glycerol, 1 mM TCEP, 1x Complete

protease inhibitor, 1x PMSF, 1x leupeptin, and 50–500 mM imidazole). Fractions were run on SDS–PAGE, stained with Coomassie blue, or immunoblotted (#14601; Abcam). Active fractions were concentrated, and buffer was exchanged with a 100-KD cut-off concentrator (Vivaspin) and stored in storage buffer (50 mM Tris–HCl, pH 8, 50 mM NaCl, 5 mM MgCl$_2$, 0.2% [mg/ml] BSA, 2 mM TCEP, 20% glycerol, and 1x protease inhibitor) at –80°C.

### In vitro Dicer cleavage assay

1 $\mu$g of T7 in vitro–transcribed substrates (vtRNAs, tRNAs, and control ncRNAs) was slow-cooled and annealed by boiling at 95°C for 3 min and then left at room temperature until temperature equilibration. Substrates were then incubated with 2 $\mu$g of recombinant purified Dicer and 1 $\mu$l of RiboLock RNase inhibitor in Dicer reaction buffer (50 mM Tris–HCl, pH 8, 300 mM NaCl, 20 mM Hepes, 5 mM MgCl$_2$, 5% glycerol, and 1x protease inhibitor), and incubated for 4 h at 37°C. Aliquots were collected at time points by mixing with 1X TBE–urea loading dye (Novex) and immediately frozen on dry ice. Samples were run on a 10% urea–polyacrylamide gel in 1x TBE at 450 V and then transferred onto Hybond-N + nylon membrane (Cytiva) for 1 h at 5 V, followed by UV crosslinking and then pre-hybridisation for 30 min in ULTRAhyb-Oligo buffer (Thermo Fisher Scientific) at 42°C. Probes were radiolabelled with 32P-ATP by polynucleotide kinase (PerkinElmer) for 1 h at 37°C, purified using a G-25 Sephadex column (GE Healthcare), and then incubated with membranes overnight. Membranes were then washed 2x with 0.1x SCC buffer and subjected to autoradiography.

### FISH for vtRNA1-2 and svtRNA1-2 detection

HEK293T, HEK293T vtRNA1-2 KO, and HeLa cells were seeded on coverslips to reach 40% confluency after 24 h. Cells were rinsed twice with PBS and fixed with PFA 4% in PBS for 15 min at 37°C. Samples were first washed twice with 1-methylimidazole solution (0.13 M 1-methylimidazole and 300 mM NaCl in DEPC H2O, pH 8) at room temperature for 10 min followed by incubation with freshly prepared EDC (1-ethyl-3-(3-dimethylaminopropyl) carbodiimide hydrochloride) solution (EDC 0.1 M in 1-methylimidazole solution at pH 8) in a humidified chamber at room temperature for 1 h. Subsequently, samples were incubated twice with 1% wt/vol glycine in PBS for 5 min at room temperature and rinsed twice with PBS. The permeabilisation step has been performed by incubating samples with 0.2% Triton in PBS for 10 min at room temperature and rinsing twice with PBS. SSC blocking buffer solution (SSC 2x, BSA 2%, sssDNA 40 ng/$\mu$l, yeast tRNA 40 ng//$\mu$l, and RNasin Plus 0.67U/$\mu$l in DEPC H2O) has been used before the probe hybridisation step. Samples were incubated in a humidified chamber at 37°C for 1 h. FISH probe buffer (SSC 2x, sssDNA 30 ng/$\mu$l, yeast tRNA 30 ng/$\mu$l, RNasin Plus 1U/$\mu$l, and 100 nM or 50 nM of fluorescently labelled probes) was added to the samples and let hybridise at 95°C for 5 min and 4°C overnight. Post-hybridisation washes (three washes at 4°C for 5 min each) were performed using different SSC buffer solutions in DEPC H2O (SSC 2x, BSA 1%; SSC 1x, BSA 1%; and SSC 0.5x, BSA 1%) and by rinsing twice with PBS.

### Identification of svtRNA in mouse tissue samples

To investigate the possible expression of murine svtRNA, sRNA-seq data were used from various mouse tissues, including spleen, liver, lungs, muscle, brain, and pancreas (GSE119661). The data were obtained in triplicates for most of the tissues and then in duplicates for brain and heart tissues, and processed according to published methods. Briefly, obtained raw reads were processed and aligned to the mouse genome index (m39) using STAR (v2.7.3a), and after the alignment, I generated normalised coverage files in the bigWig format using the bamCoverage function from deepTools (v2.2.2) and visualised them in Integrative Genomics Viewer (IGV).

### Ago1-4 PAR-CLIP analysis

The Ago1-4 PAR-CLIP dataset adapter sequences were trimmed using Cutadapt (v.1.8.3) (Marcel, 2011). Then, the reads were mapped to the reference human genome (hg38) by Bowtie (v.1.1.2) (Langmead et al, 2009), allowing for two alignment errors (mutation, insertion or deletion). For each read, only the best mapping was reported out of a maximum of 10 genomic matches. Any tag with over 10 genomic matches was discarded. After the conversion subtraction, reads that mapped to only one genomic location were retained for further analysis. The clusters of reads (Ago binding sites) were identified based on T-to-C conversions using PARalyzer (v.1.5) (Corcoran et al, 2011) detailed settings described in Table S2. Then, these files were visualised using IGV (Robinson et al, 2011; Thorvaldsdottir et al, 2013).

### Ago2/3 RIP-seq

The Ago2/3 RIP-seq datasets consisted of one replicate per Ago2/3 sample (HEK293T cell line). Next, the 3′ adapter sequence (TCTCGTATCGTATGCCGTCTTCTGCTTG) was trimmed using fastx-clipper from fastx toolkit (v.0.0.13.2) from Hannon Lab with a quality score of 33 and a read length longer than 15 nt. The index of the human genome was built with bowtie-build (v.1.1.2) (Langmead et al, 2009), and reads were mapped to the reference human genome (hg38) using Bowtie (v.1.1.2) (Langmead et al, 2009). The Stranded-Coverage tool from GitHub (https://github.com/pmenzel/stranded-coverage) was used to calculate the read coverage for vtRNA in the genome with the following settings: -f, -x, -n (normalisation of the coverage [Reads Per Million]), -s 1. Output files were visualised using IGV.

### Nuclear/cytoplasmic Ago2 RIP-seq

The reads were trimmed using cutadapt with the following settings: -e 0.05 -q 23 -m 15 -M 30 -a "A(Arteaga & Engelman)," and aligned to hg38 using STAR aligner allowing a maximum of three mismatches and retaining only uniquely mapped reads. Next bamCoverage function was used from deepTools (Ramirez et al, 2014) to calculate coverage files using −binSize 1 and −normalizeUsing RPKM into bigWig files, which were next visualised by IGV.

### Dicer PAR-CLIP analysis

FASTQ files (three biological replicates) were trimmed to remove the 3′ adapter sequence (TCTCACGTCGTATGCCGTCTTCTGCTTG,TCTC-CATTCGTATGCCGTCTTCTGCTTG,TCTCCCATCGTATGCCGTCTTCTGCTTG) and select reads longer than 15 bases in size using Cutadapt (v.1.8.3). The alignment was done using Bowtie (v.1.2.3) with the following settings: −v 3 −a −m1 −best −strata. The clusters of reads (Dicer binding sites) were identified based on T-to-C conversions using PARalyzer (v.1.5) detailed settings described in Table S2. Then, these files were visualised using IGV.

### ChRNA-seq Dicer and Ago2 KD

The adapter sequences (-A AGATCGGAAGAGCGTCGTGTAGGGAAA-GAGTGT and -a AGATCGGAAGAGCACACGTCT) were trimmed from datasets (two replicates per condition) by Cutadapt (v.1.8.3), discarding processed reads shorter than 10 nt. The processed reads were then mapped to the human reference genome (hg38) using STAR aligner (v.2.5.3a). The basic genome annotation was downloaded from the GENCODE website. The genes were quantified using featureCounts (v.1.6.2). Any gene with less than five read counts per replicate in at least two conditions was removed, and the counts' matrix was then normalised using DESeq2.

### Generation of vt1-2 KO cell line

Cas12a-gRNA ribonucleoprotein complexes containing one sgRNA targeting the vault1-2 gene (TTTAGCTCAGCGGTTACTTCGAGTACA) were nucleofected in HEK293T using Neon Transfection System (Thermo Fisher Scientific). After 48 h, cells were single-cell–sorted into 96-well plates and subsequently genotyped. Cells bearing homozygous deletion of vtRNA1-2 were confirmed by Sanger sequencing.

### MTS proliferation assay

WT and vtRNA1-2 KO cells were seeded into 96-well plates at a density of 100 cells per well and left to proliferate for 7 d. The MTS assay was carried out using CellTiter 96 AQueous One Solution Cell Proliferation Assay (MTS) (Promega) according to the manufacturer's instructions, and absorbance was measured at 490 nm.

### Colony formation assay

To assess the impact of KO of vtRNA1-2 on cell survival, colony formation assays were carried out. WT and vtRNA1-2 KO cells were seeded into 12-well plates at a density of 1,000 cells per well and left for 10 d until visible colonies formed. Plates were stained with crystal violet staining solution (0.5% crystal violet and 20% ethanol) for 30 minutes and then quantified using the ColonyArea plugin on ImageJ.

### Rescue experiment of the HEK293T vt1-2 KO cell line

HEK293T vt1-2 KO cells were seeded into six-well plates to reach 40% confluency the next day. siRNA against Dicer or non-silencing

control siRNA at a concentration of 100 nM has been transfected using Lipofectamine RNAiMAX Reagent following the manufacturer's instructions. 24 h after, cells were transfected with either pH1shluc or pH1vt1-2 or pH1svt1-2 vectors using Lipofectamine 3000 Reagent following the manufacturer's instruction. After an additional 24 h, cells were harvested and seeded into 6-well plates at a density of $3 \times 10^5$ and left to proliferate for 3 d. Finally, plates were stained with crystal violet staining solution (0.5% crystal violet and 20% ethanol) for 30 min and then quantified using ImageJ.

### Confocal and bright-light microscopy

HEK293T cells were grown on poly-L-Lys coverslips and transfected with 100 nM of single-stranded or 40 nM of double-stranded 5′-Cy3–labelled svtRNA1-2 24 h after seeding. Cells were fixed at specific time points (4, 6, and 24 h) with paraformaldehyde (4% in PBS) and subsequently imaged on an FV1200 Olympus laser scanning confocal microscope equipped with an argon laser and a He/Ne 633-nm laser, using a 60x/NA1.4 oil immersion objective. Comparative immunofluorescence analyses were performed maintaining identical acquisition parameters. Nuclear foci were analysed with CellProfiler software, and statistical analysis was performed using GraphPad Prism. Bright-light microscopy was performed using a 40x lens.

### ChrRNA-seq sample preparation

Chromatin RNA samples for sequencing were prepared according to previous publications (Nojima et al, 2016; Di Fazio et al, 2022) with some minor modifications. Briefly, WT and vtRNA1-2 KO cells were harvested from 15-cm dishes and lysed in HLB + N buffer (10 mM Tris–HCl [pH 7.5], 10 mM NaCl, 2.5 mM MgCl2, and 0.5% NP-40), and underlaid with HLB + NS buffer (10 mM Tris–HCl [pH 7.5], 10 mM NaCl, 2.5 mM MgCl2, 0.5% NP-40, and 10% sucrose), before centrifugation at 420$g$ for 5 min. Nuclear pellets were then resuspended in NUN1 buffer (20 mM Tris–HCl [pH 7.9], 75 mM NaCl, 0.5 mM EDTA, and 50% glycerol). Chromatin was extracted by addition of NUN2 buffer (20 mM Hepes–KOH [pH 7.6], 300 mM NaCl, 0.2 mM EDTA, 7.5 mM MgCl2, 1% NP-40, and 1 M urea) on ice for 15 min with vortexing. Samples were centrifuged at 17,000$g$ for 10 min and chromatin pellets collected. Chromatin was digested by the addition of Proteinase K and Turbo DNase (Thermo Fisher Scientific). RNA was extracted from the chromatin samples using TRIzol (Invitrogen) and chloroform extraction, followed by Monarch Total RNA Miniprep Kit (NEB) according to the manufacturer's instructions, and eluted in water.

### Western blotting

4x Laemmli buffer (0.2 M Tris–HCl, 8% [wt/vol] SDS, 40% glycerol, 20% [vol/vol] $\beta$-mercaptoethanol, and 0.005% bromophenol blue) was used to treat the whole-cell and chromatin extracts after 95°C incubation for 5 min and sonication. Samples were separated on mini-PROTEAN TGX gels (Bio-Rad Laboratories), followed by transfer onto nitrocellulose membrane (Protran; GE Healthcare), and probed with tubulin (55 kD) and H3 (15 kD) antibodies.

### svtRNA1-2 transfection

MCF-7 cells were seeded to reach 70% confluency at the time of transfection. 5′-phosphorylated single-stranded vtRNA1-2 at a concentration of 100 nM has been transfected using Lipofectamine RNAiMAX Reagent following the manufacturer's instructions. 24 h after, cells were harvested and total RNA was extracted using TRIzol LS Reagent.

### RT–qPCR

1 $\mu$g of total RNA was reverse-transcribed with gene-specific primers using SuperScript III following the manufacturer's instructions.

1:10 of cDNA was subsequently analysed using the SYBR Green–based qPCR assay following the manufacturer's instructions. Quantification of the nascent RNA expression level of ERBB4 and Bcl-2 genes was normalised to the GAPDH RNA nascent expression level using the 2-ΔΔCt method. All RT–qPCR experiments were performed using three biological replicates.

### Primer sequences

A list of all primers used in this study is available in Table S7.

### Plasmid construction for rescue experiment

DNA inserts bearing the human H1 promoter sequence (in bold) upstream either short hairpin luciferase (shluc) or vt1-2 (vt1-2) or small vt1-2 (svt1-2) sequences (bold) have been obtained by gBlock gene fragment synthesis (IDT) and subsequently cloned into a plasmid backbone via Gibson assembly protocol (New England Biolabs) following the manufacturer's instructions. Recombinant plasmid sequences have been verified by Sanger sequencing:

(H1shluc)

GAACGCTGACGTCATCAACCCGCTCCAAGGAATCGCGGGCCCAGTGTCAC TAGGCGGGAACACCCAGCGCGCGTGCGCCCTGGCAGGAAGATGGCTGTGAGGG ACAGGGGAGTGGCGCCCTGCAATATTTGCATGTCGCTATGTGTTCTGGGAAA TCACCATAAACGTGAAATGTCTTTGGATTTGGGAATCTTATAAGTTCTGTATGAGA CCACAGATCCCC**GGATTCCAATTCAGCGGGAGCCACCTGATGAAGCTTGACGG GTGGCTCTCGCTGAGTTGGAATCCATTTTT**;

(H1vt1-2)

GAACGCTGACGTCATCAACCCGCTCCAAGGAATCGCGGGCCCAGTGTCAC TAGGCGGGAACACCCAGCGCGCGTGCGCCCTGGCAGGAAGATGGCTGTGAG GGACAGGGGAGTGGCGCCCTGCAATATTTGCATGTCGCTATGTGTTCTGGGA AATCACCATAAACGTGAAATGTCTTTGGATTTGGGAATCTTATAAGTTCTGTAT GAGACCACAGATCCCC**GGGCTGGCTTTAGCTCAGCGGTTACTTCGAGTAC ATTGTAACCACCTCTCTGGGTGGTTCGAGACCCGCGGGTGCTTTCCAGCT CTTTTT**; and

(H1svt1-2)

GAACGCTGACGTCATCAACCCGCTCCAAGGAATCGCGGGCCCAGTGTCAC TAGGCGGGAACACCCAGCGCGCGTGCGCCCTGGCAGGAAGATGGCTGTGAG GGACAGGGGAGTGGCGCCCTGCAATATTTGCATGTCGCTATGTGTTCTGGGAAA TCACCATAAACGTGAAATGTCTTTGGATTTGGGAATCTTATAAGTTCTGTATGAG ACCACAGATCCCC**CGCGGGTGCTTTCCAGCTCTTTTT**.

### ChrRNA-seq WT and KO

The samples were prepared by isolating chromatin-associated RNA from HEK293T cells in wild-type and svtRNA1-2 KO conditions in triplicates. The reads were trimmed by Cutadapt (v.1.8.3), discarding processed reads shorter than 10 nt. The processed reads were then mapped to the human reference genome (hg38) using STAR aligner (v.2.5.3a). The basic genome annotation was downloaded from the GENCODE website. The genes were quantified using featureCounts (v.1.6.2), and the counts' matrix was then used for the differential gene expression analysis with DESeq2. The gene ontology enrichment analysis was performed using ShinyGO 0.76 (Ge et al, 2020).

### miRanda target predictions

The sequences of the identified significantly up-regulated genes in Dicer KD and Ago2 KD conditions were downloaded from BioMart (Kinsella et al, 2011). The identified svtRNA1-2 target genes were predicted by running miRanda with the following settings: -sc 150 -en −30 -strict (v.3.3a) (Enright et al, 2003). The information such as gene target name, mapped svtRNA name, its coordinates, and the scores was obtained from the miRanda output file. The annotation of the target genes was obtained from BioMart. The custom gene feature file (intron, exon, UTR5, UTR3, and UTR5/exon meaning UTR exon junction same for UTR3/exon) in the BED format was constructed from the GENCODE basic annotation file. The canonical transcript file was constructed from the GENCODE basic annotation file taking one transcript per gene with the highest number of relevant tags that represent transcript confidence. Next, we used bedtools (v.2.27.10) intersect function with selected options: -wa, -wb, -loj, to identify the features of the mapped vault RNA fragment by the intersection of predicted mapping coordinates and the genomic feature coordinates file. Then, we used Python 3 scripts to visualise the result. The percentage of predicted mappings in a particular feature was calculated by normalising the number of predicted mappings to the feature's length.

### phastCons conservation analysis

We used phastCons scores resulting from the multiple alignments of 100 vertebrate genomes to the human genome. The files were downloaded in the wigFix format from the UCSC database (hg38.100way.phastCons). We extracted the phastCons scores corresponding to vtRNA1-2 and calculated the average score. The density of the normalised phastCons scores was visualised using Python 3.

### Secondary and tertiary structures of vtRNA molecules

We used RNA immunoprecipitation pull-down selective 2′-hydroxyl acylation followed by the primer extension (icSHAPE-MaP) (Luo et al, 2021) dataset to refine vtRNA secondary and tertiary structure predictions. RNAfold (Lorenz et al, 2011) command line version with vault RNA sequences in the FASTA format and additional −shape = vault RNA icSHAPE-MaP data were used to predict secondary structures:

```
>vtRNA1-1
GGGCUGGCUUUAGCUCAGCGGUUACUUCGACAGUUCUUUAAUUGAA
ACAAGCAACCUGUCUGGGGUUGUUCGAGACCCGCGGGCGCUCUCCAGUCCUUUU
(((((((....((((.(((((...((((...(((........)))....(((((((....)))))))))))))).))))))).....)))))))....
(−58.45)
>vtRNA1-2
GGGCUGGCUUUAGCUCAGCGGUUACUUCGAGUACAUUGUAACCACCU
CUCUGGGGUGGUUCGAGACCCGCGGGUGCUUUCCAGCUCUUUU
(((((((....((((((.((((....((((.........(((((((((....)))))))))))))..)))).)).))))..)))))))....  (−53.30)
>vtRNA1-3
GGGCUGGCUUUAGCUCAGCGGUUACUUCGCGUGUCAUCAAACCACCUC
UCUGGGGUUGUUCGAGACCCGCGGGGCGCUCUCCAGCCCUCUU
(((((((....((((.((((..........(((.((.(((.((((......))).)))).))))))))))))).....)))))))....  (−58.21)
```

There were no icSHAPE-MaP data for vault RNA2-1.

Next, we used RNAComposer Automated RNA Structure 3D Modeling Server (Antczak et al, 2016) to model 3D structures of the vtRNA. The output structure was then downloaded in the .pdb format and imported into Jmol 9 (http://www.jmol.org/) for visualisation and further modifications.

## Data Availability

Information about all published datasets used in this study is summarised in Table S2. ChrRNA-seq data from WT and vtRNA1-2 KO cells are deposited and accessible in GEO with accession number GSE198298.

## Supplementary Information

## Acknowledgements

We thank all members of the Gullerova group. We also thank Alan Wainman for his help with microscopy. We acknowledge the ENCODE Consortium and the ENCODE production laboratory/laboratories for generating the dataset(s). This work was supported by the Senior Research Fellowship by Cancer Research UK (grant number BVR01170) awarded to M Gullerova, EPA Trust Fund (BVR01670) awarded to M Gullerova, and Lee Placito Fund awarded to M Gullerova. The ERBB4 target explorer was generated using QIAGEN Ingenuity Target Explorer (QIAGEN, Inc.; https://targetexplorer.ingenuity.com/). The experimental illustrations were created with BioRender.com.

### Author Contributions

A Alagia: conceptualisation, validation, investigation, visualisation, methodology, and writing—review and editing.
J Tereňová: investigation and writing—original draft.
RF Ketley: investigation, visualisation, and writing—review and editing.
A Di Fazio: investigation.
I Chelysheva: supervision, investigation, visualisation, and writing—review and editing.

M Gullerova: conceptualisation, resources, data curation, supervision, validation, investigation, visualisation, project administration, and writing—review and editing.

## Conflict of Interest Statement

The authors declare that they have no conflict of interest.

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
