## [Reviewer comments · Life Science Alliance]

Life Science Alliance

Small Vault RNA1-2 modulates expression of cell membrane proteins through nascent RNA silencing

Adele Alagia, Jana Terenova, Ruth Ketley, Arianna Di Fazio, Irina Chelysheva, and Monika Gullerova

DOI: <https://doi.org/10.26508/lsa.202302054>

Corresponding author(s): *Monika Gullerova, University of Oxford*

Review Timeline:	Submission Date:	2023-03-23
	Editorial Decision:	2023-03-23
	Revision Received:	2023-03-24
	Accepted:	2023-03-28

Scientific Editor: *Eric Sawey, PhD*

Transaction Report:

Please note that the manuscript was previously reviewed at another journal and the reports were taken into account in the decision-making process at Life Science Alliance.

Referee #1 Review

Report for Author:

Terenova and colleagues submitted a manuscript about a first report of a vault RNA 1-2-derived small RNA in a human cell line. Vault RNAs are polymerase III transcripts and have initially be described as integral part of the vault complex, a gigantic RNP particle with still largely unknown function(s). Yet, recent years have witnessed evidence that the four human vault RNA paralogs are predominantly not associated with the vault complex and likely fulfill a function on their own. Here the authors provide evidence that vault RNA 1-2 is processed by Dicer into a small 3´ fragment that associates with Argonaute 2. Furthermore they propose that this fragment together with Ago2 is involved in "nascent RNA silencing" (NRS) in the nucleus, a very recently described mechanism of gene regulation by the same research group.

The submitted study is interesting and provides novel insight into vault RNA biology. However, the manuscript in its present form suffers from over-interpretations and the lack of supporting experimental data. If the authors can provide answers to the most critical points, this study might be transformable into a convincing research article.

Major points:

- 1) Abstract (line 37); Introduction (line 101); Discussion (lines 347-348): the authors use strong words when they refer to their computational prediction that the vault RNA1-2 fragment targets nascent transcripts with a double-seeded manner. In fact the entire paragraph in the Results (lines 239-268) and Fig. 6 is solely a biocomputational prediction and by no means a mechanism of action. Without experimental validation (e.g. using reports with mutations in the predicted seed regions as well as compensatory base changes) the authors will have a hard time convincing readers in the field of their new "nascent RNA silencing" (NRS) mechanism.
- 2) along the same lines: the authors must make clear why they focused their validation to only one putative mRNA target, namely ERBB4. By combining RNA-seq data and Argonaute or Dicer PAR-CLIP data together with their chromatin-associated RNA-seq (ChrRNA-seq), they came up with a list of 26 possible targets for the vault RNA1-2-derived fragment. Yet, they solely focused in this submission on a single target for validation (ERBB4). And based on Fig. 5A, ERBB4 was not even the most significantly affected mRNA upon knock-out of vtRNA1-2.
- 3) Has ERBB4 ever been reported previously to regulate proliferation? If so a reference would be needed in the manuscript.

4) Many Figures seem to be skewed or incomplete (at least on the submitted pdf as well as on the power point file): Fig. 2A,B: labeling is missing; Fig. 3B and Fig. 5A: no data points are visible on the volcano plots; Fig. 6B: no bar plots are visible

5) Introduction (lines 42-48): For my taste the authors describe gene silencing in a way too narrow manner. I am slow to accept that there are solely RNAi and modulating heterochromatin mechanisms. In recent years many new ncRNA-based gene regulatory mechanisms have been uncovered: e.g. tRNA-derived fragments can regulate translation initiation (e.g. PMID: 21855800); RNA modification can regulate the structure and/or stability of mRNAs, etc

6) Introduction, lines 67-73: it might have escaped the authors attention that in the recent past, several studies have been published showing vtRNA functions that are independent of the vault particle. It would be scholarly to mention this here and give reference to some of those studies. (note: Horos et al. 2019 does not report on a role of the vault complex on autophagy, rather than the vault tRNA1-1 transcript).

7) For the proposed "nascent RNA silencing" (NRS) nuclear localization of the small RNA is central. Thus it would be important to know the fraction of the vtRNA1- fragment that actually localizes to the nucleus, since this fragment has also been shown by RNA-Seq to be present in the cytosol. Furthermore the transfected fluorescently labeled small vault RNA1-2 fragment is visible as distinct puncta in the nucleus (Figure 2C). Do the authors have any idea why the green fluorescence accumulates in such discrete loci? Furthermore many of these puncta actually do not localize within the nucleus (see Fig 2C, ds svtRNA1-2 at 6 hrs). The sequence of the used synthetic vtRNA1-2 fragments needs to be given in the Methods section.

8) Line 140 (and at several other places throughout the manuscript) the authors state that vault RNA and tRNA have many similarities. It is unclear what the authors actually mean by that. I think beside the polymerase III transcription and the fact that they can give rise to smaller functional fragments, these two ncRNA classes are rather distinct.

9) Line 299: a reference for the statement that vault RNAs are differentially expressed in various tissues is needed there.

10) Line 152 and at several other places within this manuscript, the authors state that tRNA fragments (tsRNA in their used nomenclature) function in the nucleus (Di Fazio et al 2022). I strongly suggest to soften and rephrase this since it could be misunderstood by readers. The majority of tRNA-fragments or tRNA halves do not function in the nucleus or in an RNAi-like manner.

11) Supplementary Fig. 1 D: the quality of the presented northern blot does not allow drawing any conclusions about the vault RNA abundance in the two cell lines. This northern needs to be repeated and improved in order to become convincing.

12) Figure 7D: the depicted model is misleading. It appears as if all vault RNA1-2 transcripts that are not associated to the vault complex get processed by Dicer. Given the fact that the vast majority of the vault RNAs never see the vault complex (~ 90%), this would imply that no free full length vtRNA1-2 would be present in the cells. In other words: what is the fraction of vault complex-free vtRNA1-2 that gets processed by Dicer?

Minor points:

a) Line 49: delete "a"

b) Line 70: the publication (Persson et al 2009 is not a review article

c) Line 67: a reference should be given there since it is unclear to which vault RNAs the authors refer to. Human vault RNAs are neither 88 nor 142 nt long, for example.

d) Abbreviations should be defined upon first mentioning: e.g. MA (line 117); IGV (Figure 2 legend)

e) Fig 7 B and C: the labeling below the data is confusing (Vault1-2), since it could be misunderstood as full length vault RNA1-2

Referee #2 Review

Report for Author:

Terenova et al.

Small non-coding vault RNA1-2 modulate expression of cell membrane proteins through nascent RNA silencing

In this manuscript, the authors have used Dicer knock down cells and find that the non-coding vault RNAs produce small, Dicer-dependent RNAs. Publicly available PAR-CLIP data revealed that small vault RNAs are found in Ago1-4 complexes suggesting functional small RNAs. The authors further report that such complexes may predominantly act in the nucleus on nascent RNAs (shown by ChrRNA-seq). The authors predict potential targets and use a number of different assays to validate a few of them. Knock out of vtRNA1-2 leads to defects in cell proliferation and the affected target genes encode for membrane proteins. Thus, it is speculated that svtRNAs regulate membrane functions. Terenova et al. further investigate a potential non-canonical binding mechanism involving 5' and 3' seeds and suggest a target looping between the two seed regions. Finally, they show that synthetic svtRNA1-2 down-regulates its target ERBB4, which is important for breast cancer cells and could be a strategy to

target this cancer.

This is a comprehensive study of small RNA species originating from vtRNAs. However, it has been reported before that vtRNAs can be processed to small RNA species and in several studies Dicer-dependency and loading into Ago has been reported. Thus, the study lacks novelty. Moreover, many conclusions are not fully justified by the data and in addition, there are a number of shortcomings and other issues with this study that are listed below.

1. Figure 1: how efficient is the Dicer knock down? How many different siRNAs have been used and how do these knock downs compare to the many studies published during the last decade attempting to find Dicer-dependent and -independent small RNA species? Since there has been so much work in this area it is surprising that the authors do not investigate this more thoroughly.
2. The red and blue lines in Figure 1B are not too different, which would be consistent with a rather mild or weak Dicer knock down. Only 40% of the affected small RNAs are miRNAs. This is also unusual since this should by far be the dominant population in these datasets.
3. Figure 1F: the vtRNA would not look like a Dicer substrate. It has a long 3' overhang, which might inhibit Dicer processing. It is known that addition of nucleotides to the 3' end of pre-miRNAs can either alter or fully abolish Dicer processing. Thus, the presented structure might not be a good substrate. Direct Dicer cleavage assays might help to show Dicer processing compared to a canonical substrate.
4. Suppl. Figure 1: first, D is of very low quality and should be removed and repeated. Two lanes are presumably on this blot and I do not understand why such an image is included in a submission. Second, 1B is somewhat puzzling. The figure indicates vtRNA1-2, 2-1 and 1-1 but I guess small RNAs derived from these RNAs are meant? In Dicer knock down cells, svtRNAs1-2 and 2-1 are downregulated. In the text, the authors state that svtRNA1-1 are unchanged and look like WT? This is not shown in this Figure. In fact, it seems to be the opposite. In Dicer knock down cells, these small RNAs are actively up-regulated? This is confusing.
5. Although less reads, the images of vtRNA1-2 and vtRNA1-1 shown in suppl. Figure 3 are very similar. However, the main idea of this study is that vtRNA1-1 is different and does not produce functional small RNAs. This needs to be clarified and investigated more thoroughly.
6. Parts of Figures 2A, B and 3F seem to be missing and thus it is really difficult to assess the shown content.
7. Figure 2: What are the foci shown in these images? Is this anything functional? Is it specific to the transfected RNA? The authors need to include a number of controls in order to show convincing results. E.g. a small RNA derived from vtRNA1-1 could be a negative control.
8. In Figures 4 and 5, the authors mainly investigate effects of vtRNA1-2 knock out. Although the authors report on a small overlap between the affected targets (how many targets would overlap with a control or random dataset?), a direct link to the Dicer-produced small RNAs is not presented. This part is therefore somewhat unconnected.
9. When analyzing targets, the authors consider only up-regulated genes because the small RNAs are produced by Dicer and bound by Ago2. This assumption is very weak.
10. Why did the authors use MiRanda? There are more recent miRNA target prediction tools based on more recent findings on miRNA-target interactions.
11. The authors should not coin the term 3' complementary seed region since it is not at all clear what it is and why this would be a seed region. Seed regions are very well defined by structural features of the Argonaute protein. Would Argonaute structures allow for the proposed looping of the target RNA? This needs to be investigated in molecular detail before drawing such conclusions.

Referee #3 Review

Report for Author:

The manuscript by Terranova et al. studies the putative functions of DICER-digested fragments derived from Vault RNAs. Vault RNAs are RNA Pol III products that form large ribonucleoprotein complexes. The authors show that, similarly to tRNAs, vault RNAs are substrates of the ribonuclease DICER to generate different kinds of small fragments collectively known as svtRNAs. Among these, svtRNA1-2 seems to be the key actor of the paper in detriment of other svtRNAs like the ones named 1-3 and 2-1. They show that 1-2 associates to argonaute 2 and they claim that it regulates expression of specific genes with consequences in membrane physiology and cancer.

We are in front of a complex paper in which evidence coming from wet bench experiments is complemented with bioinformatics

evidence from genome-wide approaches that are not always equally concluding. Unfortunately the authors did not take enough care in the preparation of the manuscript and to produce a sufficiently clear text to help the reader.

Formal problems

Main

1. Lettering Figures 2A and 2B is missing. It is impossible to interpret them.
2. Figure 3B is supposed to present two volcano plots. Panels are empty.
3. I cannot tell if Figure 3F corresponds to a metagene analysis. Neither the text nor the legend indicates it clearly.
4. I do not see the point of showing the sequencing chromatogram and the sequence itself of Supplementary Figure 5A.
5. Figure 5A is supposed to present two volcano plots. The panel is empty.
6. I am not sure if Supplementary Figure 6B is complete

I must say that I opened the merged pdf file with both the Preview and the Acrobat Pro applications and in both cases I found the above-mentioned problems.

Minor

- Figure 4A is unnecessary. CRISPR-mediated KO is widely known.

Main scientific concerns

1. Assuming that unlettered Fig. 2B supports the conclusion that svtRNA1-2 associates with argonaute proteins, unfortunately immunofluorescence images of Figure 2C do not seem to support the assertion that the svtRNA1-2 fragment is enriched in nuclear as well as in cytoplasmic AGO2 fraction. Most green dots are seen within nuclei, some in the nuclear periphery and very few in the cytoplasm.
2. The authors show that 1941 and 1813 genes are upregulated in AGO2 KD and DICER KD cells respectively. Considering protein-coding genes only, they identify 1066 genes that are upregulated in common by AGO1 and DICER knockdowns. Under these conditions they find that the highest number of predicted gene targets correspond to svtRNA1-2 (216 genes)*, compared to 1-1 (73 genes) and 1-3 (75 genes). Even if 216/1066 is a considerable fraction, the authors do not mention how many genes containing svtRNA1-2 target sites are revealed by the in silico analysis and, most importantly, how many of these target genes can be assigned to genes upregulated by AGO1 and DICER knockdowns. Without such information it is difficult to conclude about the regulatory relevance of svtRNA1-2.

*Shouldn't labels of the sets in the Venn diagrams of Fig. 3B be svtRNAs and not vtRNAs?

3. What is the biological conclusion of the different density patterns in Figure 3F? What is the consequence of being concentrated at the beginning of the genes or to be dispersed intragenically?
4. Experiments in Figure 4 assess the effects of knocking out the gene for vtRNA1-2. The authors mentioned that they are assessing svtRNA1-2. However, if I understood correctly, the KO of the vtRNA1-2 gene will also affect the functions of the Vault ribonucleoprotein complex. If so, the physiological consequences of knocking down vtRNA1-2 cannot be attributed to svtRNA1-2.
5. How many of the 450 genes downregulated and 768 genes upregulated upon vtRNA1-2 knockout coincide with those regulated by DICER/AGO2 knockdowns? This would be important to assess the roles of svtRNA1-2.
6. In view of the data presented, the assertion that vtRNA1-2 is involved in the regulation of nascent RNA of protein-coding genes that are associated with cell membrane physiology is an overstatement.
7. The authors assess the effects of transfecting synthetic svtRNA1-2 in MCF7 cells and find that it downregulates ERBB4 expression (Figure 7A). Although they use adequate RNA controls (5' shuffled and 3' shuffled), there are no controls indicating what other genes are affected by this treatment and, most importantly, whether the observed inhibition of cell proliferation (Figure 7B) is only due, totally or in part, to the downregulation of ERBB4.

March 23, 2023

RE: Life Science Alliance Manuscript #LSA-2023-02054-T

Prof. Monika Gullerova
University of Oxford
Sir William Dunn School of Pathology
South Parks Road
Oxford OX1 3RE
United Kingdom

Dear Dr. Gullerova,

Thank you for submitting your revised manuscript entitled "Small Vault RNA1-2 modulates expression of cell membrane proteins through nascent RNA silencing". We would be happy to publish your paper in Life Science Alliance pending final revisions necessary to meet our formatting guidelines.

- please add ORCID ID for corresponding author-you should have received instructions on how to do so
- please add a summary blurb/alternate abstract, a category for your manuscript, and keywords to our system
- please add the Twitter handle of your host institute/organization as well as your own or/and one of the authors in our system
- please rename your EV figures as supplementary figures and adjust the figure callouts in the text accordingly
- GEO accession GSE198298 should be made publicly accessible at this point, and please remove the reviewer token from your Data Availability statement
- we do not have a separate Conclusion section, so please incorporate any thoughts into the existing Discussion section

A. FINAL FILES:

B. MANUSCRIPT ORGANIZATION AND FORMATTING:

Sincerely,

March 28, 2023

RE: Life Science Alliance Manuscript #LSA-2023-02054-TR

Prof. Monika Gullerova
University of Oxford
Sir William Dunn School of Pathology
South Parks Road
Oxford OX1 3RE
United Kingdom

Dear Dr. Gullerova,

Thank you for submitting your Research Article entitled "Small Vault RNA1-2 modulates expression of cell membrane proteins through nascent RNA silencing". It is a pleasure to let you know that your manuscript is now accepted for publication in Life Science Alliance. Congratulations on this interesting work.

DISTRIBUTION OF MATERIALS:

Again, congratulations on a very nice paper. I hope you found the review process to be constructive and are pleased with how the manuscript was handled editorially. We look forward to future exciting submissions from your lab.

Sincerely,
